# Activation of a nucleotide-dependent RCK domain requires binding of a cation cofactor to a conserved site

**Celso M Teixeira-Duarte[1,2,3], Fátima Fonseca[1,2], João H Morais-Cabral[1,2]\***

[1]Instituto de Investigação e Inovação em Saúde (i3S), Universidade do Porto, Porto, Portugal; [2]Instituto de Biologia Molecular e Celular (IBMC), Universidade do Porto, Porto, Portugal; [3]Programa Doutoral em Biologia Molecular e Celular (MCbiology), Instituto de Ciências Biomédicas Abel Salazar (ICBAS), Universidade do Porto, Porto, Portugal

**Abstract** RCK domains regulate the activity of $K^+$ channels and transporters in eukaryotic and prokaryotic organisms by responding to ions or nucleotides. The mechanisms of RCK activation by $Ca^{2+}$ in the eukaryotic BK and bacterial MthK $K^+$ channels are well understood. However, the molecular details of activation in nucleotide-dependent RCK domains are not clear. Through a functional and structural analysis of the mechanism of ATP activation in KtrA, a RCK domain from the *B. subtilis* KtrAB cation channel, we have found that activation by nucleotide requires binding of cations to an intra-dimer interface site in the RCK dimer. In particular, divalent cations are coordinated by the γ-phosphates of bound-ATP, tethering the two subunits and stabilizing the active state conformation. Strikingly, the binding site residues are highly conserved in many different nucleotide-dependent RCK domains, indicating that divalent cations are a general cofactor in the regulatory mechanism of many nucleotide-dependent RCK domains.

**\*For correspondence:**
jcabral@ibmc.up.pt

**Competing interests:** The authors declare that no competing interests exist.

## Introduction

RCK (regulator of conductance of $K^+$) domains are conserved regulatory domains of $K^+$ channels and transporters (*Giraldez and Rothberg, 2017*). These domains regulate the function of the eukaryotic Slo $K^+$ channels (*Jiang et al., 2001*), with roles in neuronal excitability, smooth muscle contractility, hormone secretion, nociception and fertility (*Barrett et al., 1982*; *Lu et al., 2015*; *Raffaelli et al., 2004*; *Santi et al., 2010*; *Tabak et al., 2011*). RCK domains are also commonly found as regulatory domains of prokaryotic $K^+$ channels and transporters, with key roles in osmoregulation, pH homeostasis, regulation of turgor pressure and membrane potential (*Bakker and Mangerich, 1981*; *Epstein and Schultz, 1965*; *Kroll and Booth, 1981*; *Meury et al., 1985*; *Whatmore and Reed, 1990*). Attesting to the importance of these domains for $K^+$ homeostasis in prokaryotes, more than half of the identified prokaryotic $K^+$ channels contain RCK domains (*Kuo et al., 2005*).

RCK domains form dimeric or pseudo-dimeric units (*Jiang et al., 2001*; *Wu et al., 2010*; *Yuan et al., 2010*), which assemble as functional octameric rings (*Albright et al., 2006*; *Jiang et al., 2002*; *Wu et al., 2010*; *Yuan et al., 2010*); the KefC $K^+$ transporter is an exception, with a RCK domain dimer as the functional unit (*Roosild et al., 2009*). The regulatory role of RCK domains results from the rearrangement of the dimer unit upon binding of signaling molecules, which ultimately causes a conformational change in the effector membrane protein. In general, RCK domains are classified as either nucleotide- or cation-dependent (*Cao et al., 2013*).

The eukaryotic Slo $K^+$ channels are regulated by cation-dependent RCK domains: BK or Slo1 is activated by $Ca^{2+}$, Slo2 by $Na^+$ and Slo3 is pH-dependent. These channels have been the target of thorough functional analyses (*Atkinson et al., 1991*; *Dworetzky et al., 1994*; *Golowasch et al.,*

*1986*; *Kameyama et al., 1984*; *Latorre et al., 1982*; *Leonetti et al., 2012*; *Pallotta et al., 1981*; *Schreiber et al., 1998*; *Sweet and Cox, 2008*; *Xia et al., 2002*; *Xia et al., 2004*; *Yan et al., 2012*; *Yuan et al., 2003*; *Cox et al., 1997 Zhang et al., 2006*). In addition, structures of full-length Slo channels and of their RCK domains have provided direct insights on channel architecture, ligand binding and mechanism of regulation (*Hite et al., 2017*; *Hite et al., 2015*; *Leonetti et al., 2012*; *Tao et al., 2017*; *Wu et al., 2010*; *Ye et al., 2006*; *Yuan et al., 2010*). The mechanism of RCK activation in BK by $Ca^{2+}$ is particularly well understood; divalent cations bind to sites formed in the activated state of the ring, stabilizing this conformation and opening the ion channel gate (*Hite et al., 2017*).

Structures of prokaryotic RCK domain regulated channels and of RCK domains alone have also been determined (*Supplementary file 1*). Among them are the GsuK and MthK $K^+$ channels (*Jiang et al., 2002*; *Kong et al., 2012*) and the cation channels TrkHA (*Cao et al., 2013*) and KtrAB (*Diskowski et al., 2017*; *Szollosi et al., 2016*; *Vieira-Pires et al., 2013*).

MthK is another well-studied example of a $K^+$ channel with a cation-dependent RCK domain. $Ca^{2+}$ stabilizes the RCK octameric ring and activates the channel in a cooperative manner (*Dong et al., 2005*; *Pau et al., 2010*; *Ye et al., 2006*; *Zadek and Nimigean, 2006*). In contrast, $H^+$ binding disrupts the RCK octameric ring (*Dong et al., 2005*; *Ye et al., 2006*) and inhibits activation of the channel (*Pau et al., 2010*). Three distinct $Ca^{2+}$ binding sites have been identified in each MthK RCK domain, six sites per dimeric unit. All sites are involved in channel activation, and some are allosterically coupled (*Pau et al., 2011*; *Smith et al., 2013*). Like in the BK channel, some of the $Ca^{2+}$ sites are at protein interfaces so that cation binding alters the conformation of the RCK dimeric unit, stabilizing the ring in an activated state and opening the channel (*Smith al., 2012*; *Smith al., 2013*).

Unlike cation-dependent RCK domains, the molecular mechanisms that underlie ligand-dependent conformational changes in nucleotide-activated RCK domains are not well understood. KtrAB is a cation channel with a nucleotide-dependent RCK domain. This channel is an essential component of the $K^+$ homeostasis machinery in many bacteria, involved in adaptation to osmotic stress and in pH regulation (*Holtmann et al., 2003*; *Ochrombel et al., 2011*). The structure of the KtrAB complex revealed a homodimeric membrane protein (KtrB) assembled with a cytosolic RCK octameric ring, formed by the KtrA protein (*Diskowski et al., 2017*; *Vieira-Pires et al., 2013*). KtrA binds nucleotides, ATP or ADP. ATP binding induces the adoption of a square-conformation by the octameric ring and increased $K^+$ flux activity through KtrAB; ADP binding results in a non-square conformation and decreased flux (*Figure 1a*). However, it is still not clear how ATP binding induces the activated conformation of the RCK ring.

In this study, we explored the mechanism of nucleotide-dependent activation in KtrA. In particular, we have identified residues in the intra-dimer interface of KtrA that are vital for activation in the presence of ATP and have demonstrated the role of these residues. We reveal that one of these residues is highly conserved and has a crucial role in cation binding especially, in the chelation of divalent cations. This role is essential for the adoption of the active conformation of the RCK domain. With this study, we have provided a new understanding of the mechanism of regulation by nucleotides in nucleotide-dependent RCK domains.

## Results

### Impact of intra-dimer interface mutants on KtrA structure and function

A comparison of the KtrA structures with ATP (PDB: 4J90) and ADP (PDB: 4J91) shows the rings in different conformations: square and non-square, respectively (*Figure 1a and b*). As previously described (*Albright et al., 2006*; *Rocha et al., 2019*; *Vieira-Pires et al., 2013*), the adoption of the square conformation upon ATP binding involves a change in the intra-dimer hinge angle (*Figure 1— figure supplement 1*). This change brings closer together the two nucleotide-binding sites in the RCK dimeric units, as seen in the D36-D36 Cα intra-dimer distance, a conserved residue in the nucleotide binding site (*Figure 1—figure supplement 1* and *Supplementary file 2*). Narrowing of the intra-dimer space with bound ATP allows the establishment of interactions between the phosphate groups and residues in both the same and opposite subunits of the KtrA dimer (*Figure 1a and b*). In particular, the R16 residues are positioned between the two nucleotides and appear to stabilize the

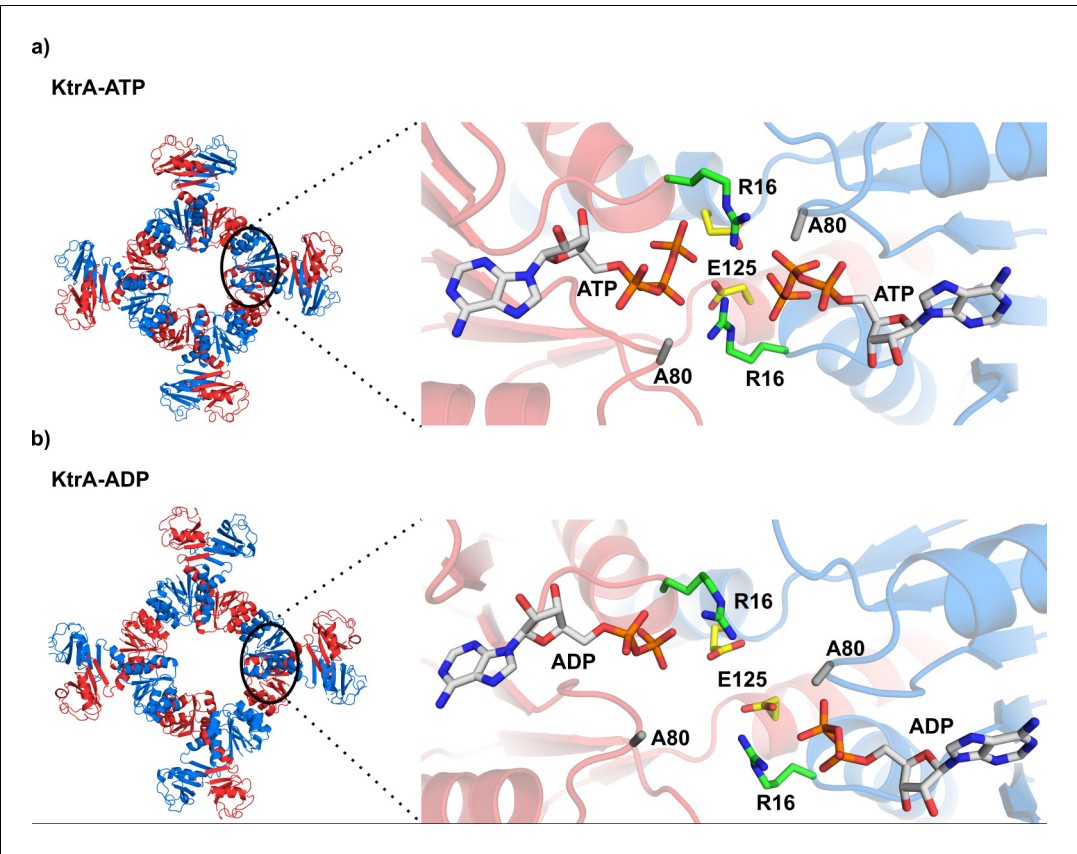

**Figure 1.** KtrA bound to ATP and ADP. (**a**) Left - square conformation of the KtrA octameric ring bound to ATP. Right- view of the ATP binding pockets in a KtrA dimer showing dimer subunits and nucleotides positioned close together. (**b**) Left- non-square conformation of the KtrA octameric ring bound to ADP. Right - view of the ADP binding pockets showing dimer subunits and nucleotides positioned farther apart relative to a). Intra-dimer interface residues discussed in the main text are shown as stick and labeled. KtrA octameric ring subunits are colored alternately blue and red. See also *Figure 1—figure supplement 1*.

The online version of this article includes the following figure supplement(s) for figure 1:

**Figure supplement 1.** Intra-dimer hinge angle in wild-type KtrA.

closely positioned negatively-charged phosphate groups. In contrast, in the ADP-bound structure the nucleotide molecules do not establish interactions with opposing subunit residues. Based on this evidence, we previously proposed that bridging interactions mediated by R16 and ATP stabilize the square or high-flux conformation of the KtrA octameric ring (*Vieira-Pires et al., 2013*).

In order to explore the functional role of R16 and E125, another residue in the dimer interface and close to the ATP phosphate groups (*Figure 1a and b*), we used an *E. coli* phenotype complementation assay. As a control, we also analyzed the impact of mutating A80 since substitution with a bulkier side-chain was expected to cause a distortion of the nucleotide-binding site. In the complementation assay, low-level constitutive expression of an active potassium channel in the $K^+$-transport deficient-strain TK2420 rescues bacterial growth in a medium containing low amounts of $K^+$ (*Albright et al., 2007*; *Buurman et al., 2004*). While empty vector-containing cells require a minimum of 30 mM $K^+$, expression of wild-type KtrAB allows growth of TK2420 in only 1 mM $K^+$ (*Figure 2*). We introduced the single-point mutations R16A, R16K, A80P and E125Q in KtrA and tested their impact in the complementation assay. As expected, A80P resulted in KtrAB inactivation. However, and contrary to our predictions, none of the R16 mutations had an impact on the growth phenotype, while E125Q resulted in an apparently inactive KtrAB (*Figure 2*).

To understand the basis of these results we first verified that the KtrA mutants A80P and E125Q retained the same basic properties of the wild-type protein. Both mutants assemble as octamers and

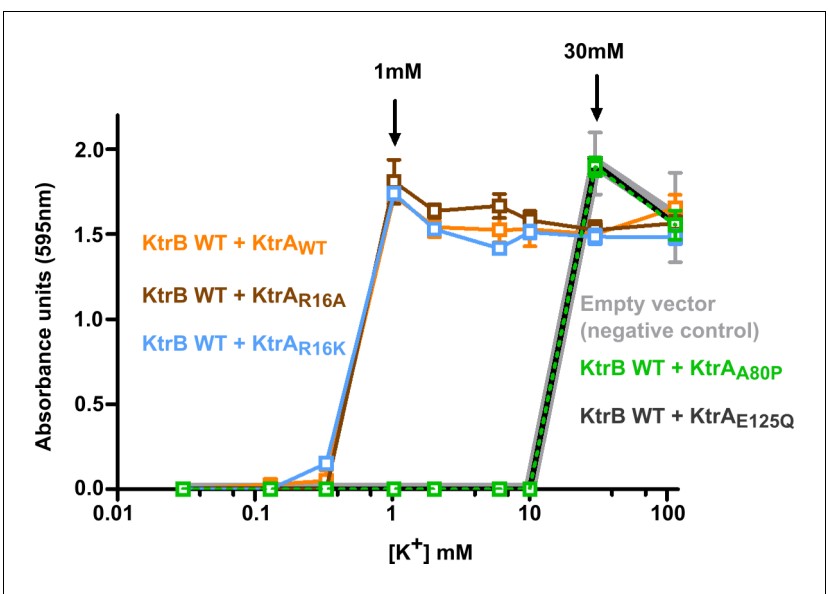

**Figure 2.** Functional impact of KtrA intra-dimer interface residues. Optical density of overnight *E. coli* TK2420 cultures (expressing different KtrAB constructs) as a function of K$^+$ concentration in the growth media. Mean ± SD (n = 6) from two independent experiments. See also *Figure 2—figure supplement 1*.
The online version of this article includes the following figure supplement(s) for figure 2:

**Figure supplement 1.** Size-exclusion chromatography elution profiles of KtrA and KtrAB proteins.

are able to form a stable complex with KtrB, as revealed by size-exclusion chromatography (*Figure 2—figure supplement 1a and b*). In addition, we determined the structures of all KtrA mutants with bound ATP or ADP (*Table 1*). The structures of R16K (at 2.7 Å resolution) and R16A (at 3.4 Å resolution) with ATP are very similar to that of the previously determined wild-type KtrA-ATP (WT-ATP) (*Figure 3a* and *Figure 3—figure supplement 1*), adopting the square conformation. A simple indicator of ring conformation is the pair of distances (L1 and L2) separating the Cα of N38 residues located in opposite subunits across a ring face (*Figure 3a* and *Supplementary file 3*). In the ATP-bound rings, these distances are similar in the R16K (30.8/30.8 Å), R16A (29.9/29.9 Å) mutants and wild-type (30.0/30.0 Å) structures. Like in the wild-type protein, the ADP-bound R16K (at 3.0 Å resolution) and R16A (at 3.7 Å resolution) rings adopt a non-square conformation (*Figure 3a*). However, the ADP-bound ring conformations are different from each other and from the wild-type, with L1/L2 distances for the wild-type of 40.7/30.7 Å and for R16K of 52.0/21.9 Å (*Supplementary file 3*). In the ADP-bound R16A mutant there are eight KtrA dimers present in the asymmetric unit, corresponding to two different rings with L1/L2 distances of 34/30 Å and 37/28 Å.

Crystal structures of the mutants that did not rescue the TK2420 growth phenotype (A80P and E125Q) reveal a different picture (*Figure 3b*). The ATP-bound structures do not adopt a square conformation, displaying L1/L2 distances for A80P (at 3.9 Å resolution) of 52/23 Å and for E125Q (at 4.1 Å resolution and with two different half-rings in the asymmetric unit) of 34/30 Å and 42/27 Å. The ADP-bound structures are also non-square, showing ring conformations that are very similar to those observed with ATP: A80P (at 4.3 Å resolution) has L1/L2 distances of 52/22 Å and E125Q (at 4.0 Å resolution and with two rings) of 34/30 Å and 42/27 Å.

It is apparent that there is a large structural variability between rings adopting the non-square conformation, as shown in D36-D36 distances and L1/L2 distances (*Supplementary file 2* and *Supplementary file 3*), allowing the clustering of these structures in four groups (*Supplementary file 3*). It is worthwhile pointing out, that besides variations between rings we also found variation in the D36-D36 distance within individual non-square rings (*Supplementary file 2*).

This structural analysis fits well with our phenotype rescue assay. KtrA mutants that show a wild-type protein function (R16K and R16A) are able to adopt a square conformation with ATP. In contrast, mutants that do not rescue growth (A80P and E125Q) also do not adopt the square

**Table 1.** Data collection and refinement statistics.

**Data collection statistics**

| | | | | |
|---|---|---|---|---|
| PDB ID | 6S2J | 6S5B | 6S5D | 6S7R |
| Crystal | R16K_ATP | R16K_ADP | R16A_ATP | R16A_ADP |
| Space group | I4 | $P2_12_12_1$ | I4 | P1 |
| Unit-cell dimensions a, b, c (Å) α, β, γ (°) | 123.3, 123.3, 84.4 90, 90, 90 | 85.8, 137.4, 202.6 90, 90, 90 | 121.6, 121.6, 83.7 90, 90, 90 | 93.0, 97.9, 144.6 90.4, 97.1, 110.3 |
| Protomers in the asymmetric unit | 2 | 8 | 2 | 16 |
| Wavelength (Å) | 0.954240 | 0.979480 | 0.978570 | 0.978570 |
| Resolution (Å) | 46.16–2.67 (2.80–2.67) | 49.51–3.05 (3.16–3.05) | 45.60–3.39 (3.66–3.39) | 48.80–3.73 (3.86–3.73) |
| No. of reflections (measured/unique) | 85203/17872 | 383771/46032 | 76213/8535 | 161349/46426 |
| Multiplicity | 4.8 (4.5) | 8.3 (4.2) | 8.9 (8.5) | 3.5 (3.0) |
| Completeness (%) | 99.0 (96.5) | 99.2 (91.6) | 99.8 (99.0) | 94.0 (51.2) |
| Rmerge (all I) (%) | 5.5 (86.8) | 14.8 (47.4) | 4.7 (182.9) | 9.5 (96.6) |
| Rmeasure (all I) (%) | 6.2 (98.1) | 15.8 (53.8) | 5.0 (194.6) | 11.2 (116.0) |
| Mean I / $\sigma$(I) | 13.9 (1.7) | 15.9 (0.2) | 17.1 (1.3) | 7.4 (1.1) |
| $CC_{1/2}$ | 0.999 (0.541) | 0.995 (0.886) | 1.000 (0.573) | 0.996 (0.494) |
| **Refinement statistics** | | | | |
| Resolution range (Å) | 46.16–2.67 | 48.17–3.05 | 45.60–3.39 | 48.80–3.73 |
| $R_{work}$/$R_{free}$ (%) | 18.96/23.57 | 20.93/24.29 | 23.83/27.90 | 24.57/30.04 |
| No. of reflections | 17870 | 46032 | 8535 | 46426 |
| Total No. of atoms | 6969 | 13784 | 3459 | 25028 |
| No. of waters | 2 | - | 2 | - |
| No. of ATP | 2 | - | 2 | - |
| No. of ADP | - | 8 | - | 16 |
| No. of Mg | 1 | - | 1 | - |
| Wilson B factor(Å²) | 90.2 | 97.4 | 194.2 | 152.9 |
| r.m.s.d. bond lengths (Å) | 0.013 | 0.003 | 0.003 | 0.003 |
| r.m.s.d. bond angles (°) | 1.086 | 0.760 | 0.745 | 0.824 |
| **Ramachandran plot** | | | | |
| Residues in favored/allowed regions (%) | 98/2 | 99/1 | 97/3 | 98/2 |
| **Data collection statistics** | | | | |
| Crystal | E125Q_ATP | E125Q_ADP | A80P_ATP | A80P_ADP |
| PDB ID | 6S5N | 6S5O | 6S5E | 6S5G |
| Space group | $C222_1$ | $C222_1$ | $P2_12_12_1$ | $P2_12_12_1$ |
| Unit-cell dimensions a, b, c (Å) α, β, γ (°) | 109.1, 156.5, 286.4 90, 90, 90 | 108.2, 155.2, 285.9 90, 90, 90 | 85.7, 133.9, 205.1 90, 90, 90 | 85.3, 132.9, 203.3 90, 90, 90 |
| Protomers in the asymmetric unit | 8 | 8 | 8 | 8 |
| Wavelength (Å) | 0.978570 | 0.978570 | 0.978570 | 0.978570 |
| Resolution (Å) | 48.24–4.09 (4.48–4.09) | 48.07–3.98 (4.36–3.98) | 49.63–3.89 (4.20–3.89) | 49.28–4.33 (4.84–4.33) |
| No. of reflections (measured/unique) | 131430/19704 | 274868/20970 | 146620/22230 | 100662/14674 |
| Multiplicity | 6.7 (6.8) | 13.1 (13.0) | 6.6 (6.6) | 6.9 (4.7) |
| Completeness (%) | 99.5 (98.3) | 99.4 (97.9) | 99.5 (98.0) | 91.9 (79.1) |
| Rmerge (all I) (%) | 12.7 (113.5) | 11.2 (132.6) | 8.1 (133.9) | 19.1 (56.8) |
| Rmeasure (all I) (%) | 13.7 (122.9) | 11.7 (138.1) | 8.8 (145.4) | 20.5 (62.9) |

*Table 1 continued on next page*

*Table 1 continued*

**Data collection statistics**

| | | | | |
|---|---|---|---|---|
| Mean I / σ(I) | 9.5 (1.7) | 12.0 (1.9) | 10.3 (1.3) | 7.9 (2.6) |
| $CC_{1/2}$ | 0.999 (0.835) | 0.999 (0.771) | 0.999 (0.830) | 0.995 (0.873) |
| **Refinement statistics** | | | | |
| Resolution range (Å) | 48.24–4.09 | 47.66–3.98 | 47.88–3.89 | 49.28–4.33 |
| $R_{work}/R_{free}$ (%) | 27.91/31.99 | 24.96/28.62 | 26.02/31.72 | 25.37/31.43 |
| No. of reflections | 19704 | 20970 | 22230 | 14674 |
| Total No. of atoms | 12594 | 12426 | 13912 | 13744 |
| No. of waters | - | - | - | - |
| No. of ATP | 8 | - | 8 | - |
| No. of ADP | - | 8 | - | 8 |
| No. of Mg | - | - | - | - |
| Wilson B factor(Å$^2$) | 183.9 | 201.8 | 188.6 | 135.7 |
| r.m.s.d. bond lengths (Å) | 0.003 | 0.003 | 0.003 | 0.003 |
| r.m.s.d. bond angles (°) | 0.804 | 0.791 | 0.883 | 0.729 |
| **Ramachandran plot** | | | | |
| Residues in favored/allowed regions (%) | 98/2 | 97/3 | 98/2 | 97/3 |

**Data collection statistics**

| | |
|---|---|
| Crystal | WT_ATP_Ca |
| PDB ID | 6S5C |
| Space group | I4 |
| Unit-cell dimensions a, b, c (Å) α, β, γ (°) | 122.7, 122.7, 84.0 90, 90, 90 |
| Protomers in the asymmetric unit | 2 |
| Wavelength (Å) | 0.978570 |
| Resolution (Å) | 45.94–3.00 (3.18–3.00) |
| No. of reflections (measured/unique) | 123606/12614 |
| Multiplicity | 9.8 (9.8) |
| Completeness (%) | 99.8 (98.9) |
| Rmerge (all I) (%) | 4.3 (141.8) |
| Rmeasure (all I) (%) | 4.6 (149.6) |
| Mean I / σ(I) | 24.5 (1.6) |
| $CC_{1/2}$ | 0.999 (0.658) |
| **Refinement statistics** | |
| Resolution range (Å) | 37.81–3.00 |
| $R_{work}/R_{free}$ (%) | 20.70/25.88 |
| No. of reflections | 12614 |
| Total No. of atoms | 3475 |
| No. of waters | - |
| No. of ATP | 2 |
| No. of ADP | - |
| No. of Ca | 1 |
| Wilson B factor(Å$^2$) | 135.1 |
| r.m.s.d. bond lengths (Å) | 0.004 |
| r.m.s.d. bond angles (°) | 0.697 |

*Table 1 continued on next page*

*Table 1 continued*

**Data collection statistics**

**Ramachandran plot**

| | |
|---|---|
| Residues in favored/allowed regions (%) | 98/2 |

Rmsd: root-mean-square deviation; values in parenthesis correspond to the highest resolution shell.

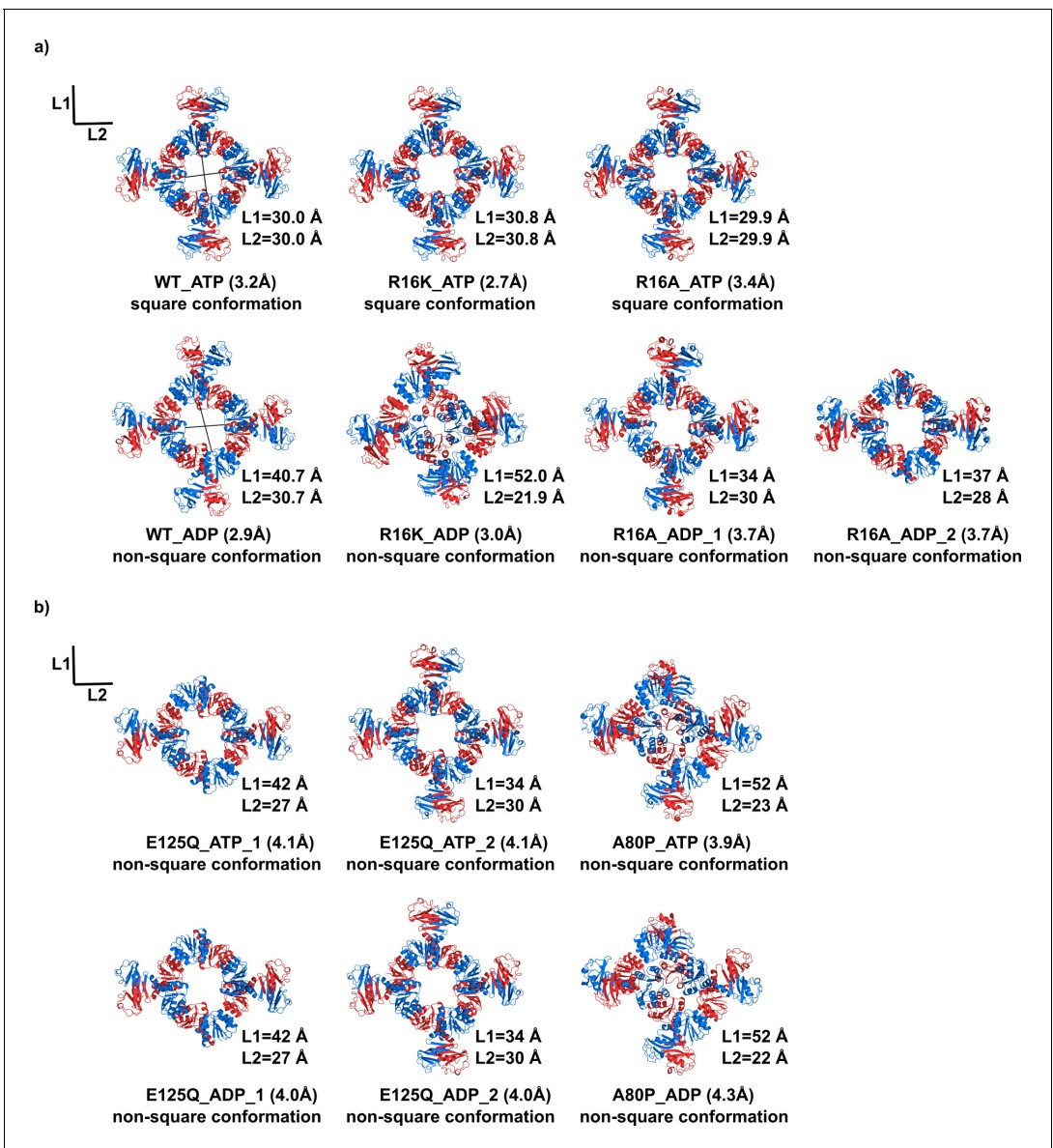

**Figure 3.** KtrA ring structures. Views of crystal structures of wild-type and mutant KtrA that are (**a**) functional or (**b**) not functional in the TK2420 phenotype rescue assay. Top rows show structures with bound ATP and bottom rows show structures with bound ADP. KtrA octameric rings are displayed with subunits colored alternately blue and red. L1 and L2 are the pair of distances between the Cα of N38 residues located in opposite subunits across the ring. In three of these rings (R16A_ADP_2, E125Q_ATP_1 and E125Q_ADP_1), poor electron-density prevented modelling of C-terminal domains. See also *Figure 3—figure supplement 1*.

The online version of this article includes the following figure supplement(s) for figure 3:

**Figure supplement 1.** Square conformation in KtrA mutants.

conformation in the ATP-bound structures. Overall, these results confirm that KtrAB activation requires the adoption of a stable KtrA-ATP ring square conformation. Importantly, they also reveal that, while R16 does not play a major role in the mechanism, A80 and E125 are crucial residues in the adoption of this conformation.

## Impact of A80P and E125Q mutations on the nucleotide-binding site

To understand the molecular basis of the functional impact of A80P and E125Q, we analyzed the nucleotide binding sites in these mutant ATP-bound structures. In this analysis, we took in consideration the low resolution of the structures and focused only on the impact of the mutations in the main-chain trace. It is worthwhile explaining that refinement of the low-resolution structures (worse than 3.5 Å) involved only rigid-body refinement of the molecular replacement model (wild-type KtrA), followed by all-atom refinement with non-crystallographic and secondary structure restraints. We performed minimal manual adjustments.

In the A80P-ATP structure, the main-chain atoms of residue P80 are positioned outside the 2Fo-Fc and Fo-Fc electron-density maps in all eight proteins in the asymmetric unit (*Figure 4*). Importantly, this offset was not the result of the refinement procedure since the position of the residue is very similar to the one in the wild-type structure. It is obvious from the maps that adjustment of the main-chain position of the residue into the density would impinge on the ATP molecule, requiring a shift in the nucleotide position to avoid steric-clashes. Unfortunately, at this low resolution it is not clear how to adjust the position of the nucleotide in the electron-density map. We concluded therefore that the A80P mutation causes a distortion of the main-chain and leads to an alteration in the position of the ATP, impeding the adoption of the square conformation.

In contrast, in the maps for the E125Q-ATP structure we do not observe obvious changes in the main-chain trace that might explain the impact of the mutation in the conformation of the ring.

## Magnesium binding site

A clue for the role of E125 is revealed by the 2.7 Å structure of the ATP-bound R16K KtrA mutant. In this structure, there was a distinct additional electron-density in the KtrA intra-dimer interface, between the γ-phosphates of the nucleotide (*Figure 5a*). After careful crystallographic refinement, we concluded that the density likely corresponds to a magnesium ion (*Figure 5a and b*). The ion is coordinated by six oxygen atoms, adopting octahedral geometry. The coordinating atoms are provided by the γ-phosphate of both ATP molecules, by two water molecules and by the carboxylic groups of the two E125 residues. In this structure, the K16 side-chain is well defined in the electron density, interacting with one of the $Mg^{2+}$ coordinating water molecules, with the carbonyl group of Q105 from the opposite subunit and with the γ-phosphate from the ATP in the same subunit. The octahedral coordination of the ion along with the majority of the coordination distances fit well with the expected coordination of a magnesium ion; however, the interaction distances with the oxygen atoms in the phosphate groups are longer than expected (*Bock et al., 1994*; *Harding, 2001*; *Zheng et al., 2008*). Calcium could also be a possible solution for the density but calcium ions are often coordinated by seven oxygen atoms in crystal structures (*Harding, 2001*), acquiring preferentially a pentagonal bipyramidal geometry according to the MetalPDB server (http://metalweb.cerm. unifi.it). In addition, the ideal coordination distances established by calcium ions are longer than for $Mg^{2+}$ (*Zheng et al., 2008*). Refinement of $Ca^{2+}$ at the site showed the same interactions

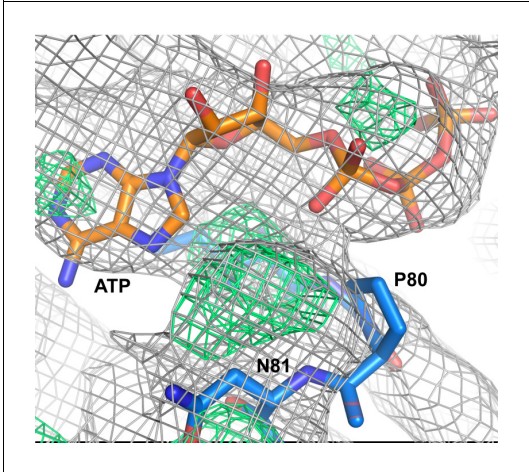

**Figure 4.** Nucleotide-binding site of the ATP-bound KtrA A80P mutant. Electron density maps 2Fo-Fc (in gray) and Fo-Fc (in green), contoured at 1.0 and 3.0σ, respectively, are shown in mesh. ATP and two residues are shown as sticks. Residue P80 is outside the electron density in the eight KtrA subunits in the asymmetric unit.

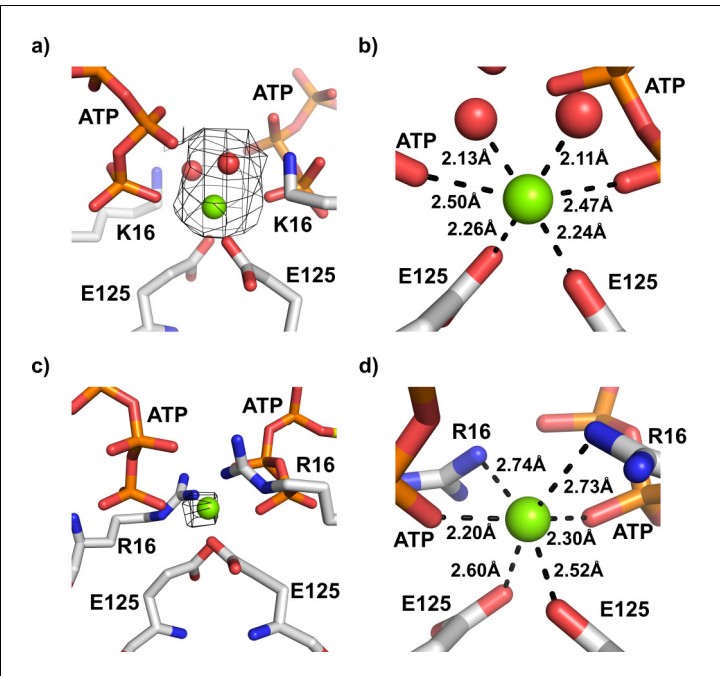

**Figure 5.** Magnesium ion binding site in the interface of KtrA dimers. (**a**) View of the intra-dimer interface of R16K-ATP with unexplained electron density in Fo-Fc map (contoured at 3.0σ and calculated at a refinement stage when the structural model did not yet include $Mg^{2+}$ or coordination waters) superposed on final refined model. (**b**) Close-up of the magnesium binding site depicted in a), showing the octahedral geometry of cation coordination. (**c**) View of the intra-dimer interface of wild-type KtrA-ATP (PDB:4J90) with Fo-Fc electron density (contoured at 3.0σ and calculated from deposited structure which does not include $Mg^{2+}$) superposed on refined model. (**d**) Close-up of the magnesium binding site shown in c), highlighting the interactions established by the cation. Magnesium ion and water molecules are shown as green and red spheres, respectively. Residues and nucleotides coordinating ions are shown as stick with coordination distances indicated.

and very similar distances to $Mg^{2+}$. Importantly, the temperature factor of $Ca^{2+}$ became larger than for the atoms forming its coordination shell (*Supplementary file 4*), suggesting that the density corresponds to a less electron-dense cation. Overall, we conclude that the ion bound to the intra-dimer interface site is a magnesium ion. It is worthwhile mentioning that $Mg^{2+}$ was not added to the protein or crystallization mixture and so the ion was sequestered during protein preparation or crystallization.

We revisited the electron-density maps of the previously resolved wild-type KtrA structure bound to ATP (at 3.2 Å, PDB code: 4J90), which also adopts a square-conformation, looking for unexplained density in the intra-dimer interface. It is possible to detect density that could also be accounted by a coordinated $Mg^{2+}$ (*Figure 5c*). Re-refinement of the structure with $Mg^{2+}$ does not cause great adjustments in the intra-dimer interface. The cation has a similar octahedral coordination to that seen in the R16K mutant structure, interacting with the oxygen atoms in the γ-phosphate of both ATP molecules and the carboxylic groups of the E125 residues. However, instead of water molecules, nitrogen atoms of the R16 guanidinium group appear to complete the coordination shell of the divalent cation (*Figure 5d*). Magnesium coordination by an arginine side-chain has been reported for at least two high-resolution structures (better than 2.0 Å) with PDB codes 1NUW and 3OS4.

Overall, these structural results reveal that E125 coordinates $Mg^{2+}$ at the intra-dimer interface. Importantly, the γ-phosphates of the ATP molecules complete the coordination of the divalent cation. In this arrangement, the divalent cation bridges the ligand molecules in two KtrA subunits, suggesting that the cation stabilizes the two subunits in the RCK dimers close together, favoring the adoption of the square conformation. Mutation to glutamine likely affects the coordination of the

$Mg^{2+}$ binding site, changing the stability of the square conformation and explaining the inability of the E125Q KtrA mutant to rescue the TK2420 phenotype.

## Functional effect of magnesium ion in KtrAB

To establish the importance of the divalent cation in the function of KtrAB we adapted the ACMA-liposome $K^+$ flux assay (*Su et al., 2016*). In this assay, KtrAB is reconstituted into liposomes in the presence of 150 mM KCl and a $K^+$ gradient is established by dilution of the proteoliposomes into a low $K^+$ concentration solution. The assay is initiated by addition of carbonyl cyanide *m*-chlorophenyl-hydrazone (CCCP, a $H^+$-ionophore), which builds-up a $H^+$ gradient across the membrane as potassium ions efflux through KtrAB. Formation of an $H^+$ gradient results in quenching of 9-amino-6-chloro-2-methoxyacridine (ACMA) fluorescence (*Figure 6*). We have found that using 2 µM ACMA in the flux assay, as others have described, causes inhibition of KtrAB. We have therefore reduced the concentration of fluorescence probe to 550 nM. In these conditions, we observe a fluorescence quenching window of ~80% in our functional assays. One- or two-phase exponential equations were fitted to the fluorescence quenching curves and flux rate constants were determined as the inverse of time constants extracted from the fit (*Figure 6—figure supplement 1* and *Supplementary file 5*). The fast component of the majority of double-exponential fits was the dominant term (larger amplitude) and is used in the analysis below. Exceptions are noted in the figure legends. In many of the experiments below we compared the activity of proteoliposomes reconstituted with different protein combinations, for example KtrB assembled with KtrA-ATP, KtrA-ADP or E125Q KtrA with bound ATP. Reconstitutions with different assemblies show similar amounts of protein associated with liposomes (*Figure 6—figure supplement 2*).

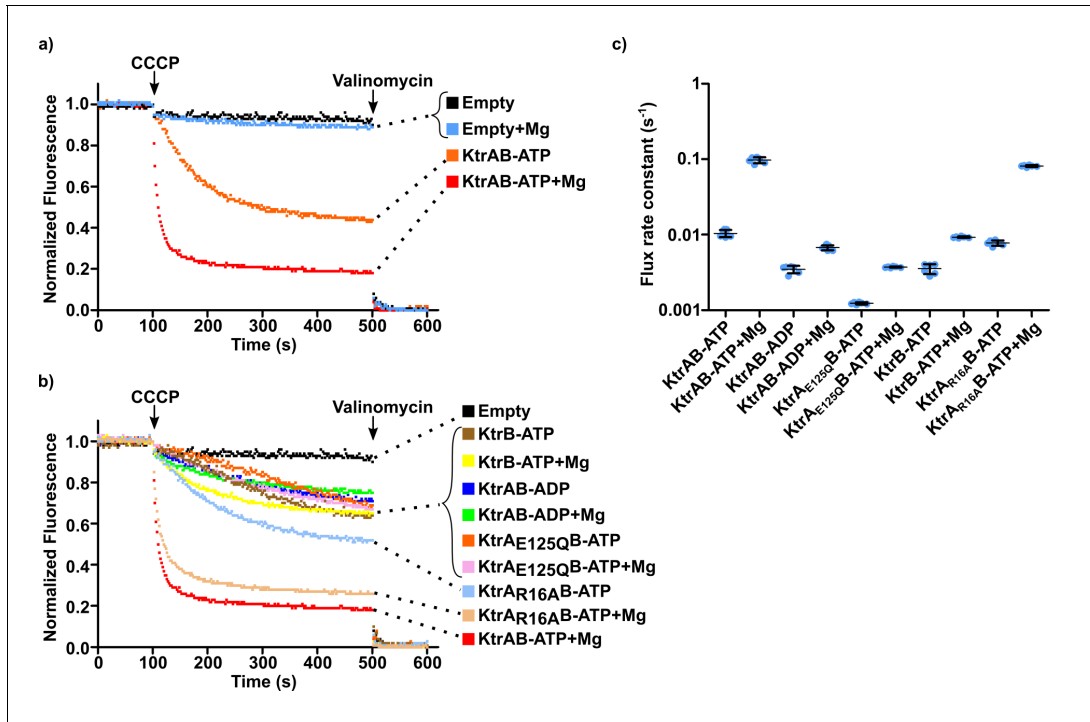

**Figure 6.** Impact of magnesium ion on KtrAB activity. (a and b) Proteoliposome flux assay performed with KtrB and wild-type and mutant KtrAB with ATP or ADP, in the presence or absence of 2 mM magnesium, as indicated. All assays were performed with sorbitol in the external solution. Panels show representative fluorescence quenching curves. Empty liposomes were used as control. (c) Plot of flux rate constants for conditions in a) and b). Mean ± SD as well as individual rate constants (blue dots) are shown for n = 6–7 from at least two different proteoliposome preparations. See also *Figure 6—figure supplement 1*, *Figure 6—figure supplement 2* and *Figure 6—source data 1*.

The online version of this article includes the following source data and figure supplement(s) for figure 6:

**Source data 1.** Flux rate constants shown in *Figure 6c*.
**Figure supplement 1.** Examples of fluorescence quenching fits with exponential equations.
**Figure supplement 2.** SDS-PAGE gel analysis of proteoliposomes.

We first compared the differences in the flux of KtrAB-ATP with and without $Mg^{2+}$, in proteoliposomes resuspended in an external solution containing 150 mM sorbitol and 1.5 mM KCl (*Figure 6a*). Flux is ~10 fold faster in the presence of $Mg^{2+}$, with the rate constant increasing from 0.011 ± 0.001 to 0.096 ± 0.008 $s^{-1}$ (*Figure 6c* and *Figure 6—source data 1*).

Next, we removed the chemical groups that coordinate $Mg^{2+}$ in the R16K and WT-ATP structures (*Figure 5b and d*) by substituting ATP for ADP or mutating E125 to a glutamine and measured flux (*Figure 6b*). Relative to KtrAB-ATP with $Mg^{2+}$, flux was 14-fold and 24-fold slower for KtrAB-ADP and $KtrA_{E125Q}B$-ATP with $Mg^{2+}$, respectively (*Figure 6c* and *Figure 6—source data 1*). For the same protein forms without divalent cation, the impact of $Mg^{2+}$ on KtrAB-ADP and $KtrA_{E125Q}B$-ATP is only a 2- to 4-fold rate increase. This difference is also observed with KtrB (*Figure 6b and c* and *Figure 6—source data 1*), indicating that the $Mg^{2+}$ effect in the ADP and E125Q protein forms is most likely not related to the mechanism of RCK activation. Like for wild-type KtrAB-ATP, $Mg^{2+}$ also activates the complex with the R16A KtrA mutant (*Figure 6b and c*, *Figure 6—source data 1*). Despite the clear impact of $Mg^{2+}$, it is noticeable that absence of added divalent cation did not abolish KtrAB-ATP activity and that its flux is larger than for KtrB or KtrAB-ADP (*Figure 6a and c*). This observation is explained in the discussion.

These functional data support the proposal that $Mg^{2+}$ is a cofactor in the nucleotide-dependent activation of KtrAB by binding at the intra-dimer RCK interface site.

## Calcium also activates KtrAB

To understand if the intra-dimer interface site is selective for $Mg^{2+}$, we performed flux experiments with addition of $Ca^{2+}$ and $Mg^{2+}$ to an external solution containing 20 mM choline chloride and sorbitol (*Figure 7*). We chose to include choline chloride because we wanted to minimize any non-specific effect of divalent cations on the liposomes by increasing the ionic strength of the assay solution and because choline is an organic cation that does not permeate KtrAB.

Relative to the condition without divalent cation, adding 0.7 or 2 mM $Ca^{2+}$ or $Mg^{2+}$ increases flux up to 4-fold. Rate constants become similar to those measured in the sorbitol/$Mg^{2+}$ condition (0.088 ± 0.0002 $s^{-1}$ for 2 mM $Mg^{2+}$ and 0.092 ± 0.001 $s^{-1}$ for 2 mM $Ca^{2+}$) (*Figure 7a and b* and *Figure 7—source data 1*). Addition of 5 mM $Ca^{2+}$ or $Mg^{2+}$ resulted in a reduction in steady-state fluorescence at 500 s to ~60% for $Mg^{2+}$ and ~35% for $Ca^{2+}$ (*Figure 7a*). No changes in total fluorescence quenching were observed when valinomycin is added to these conditions, indicating that the high concentrations of divalent cations are not affecting the structure of the proteoliposomes. However, while the flux rate constant with 5 mM $Mg^{2+}$ is unchanged, it decreases 9-fold with 5 mM $Ca^{2+}$ (*Figure 7b* and *Figure 7—source data 1*), suggesting that, at high concentrations, $Ca^{2+}$ inhibits flux by blocking the KtrAB channel.

In addition, we crystallized wild-type KtrA-ATP in the presence of 5 mM of $Ca^{2+}$ and determined the structure at 3.0 Å resolution (*Figure 7c* and *Table 1*). The structure shows an octameric ring adopting a square conformation with L1/L2 distances of 30.5/30.5 Å that are similar to those of the WT-ATP structure (*Supplementary file 3* and *Figure 3—figure supplement 1*).

The intra-dimer interface shows a bound calcium ion (*Figure 7d*). After refinement, we defined a coordination shell formed by oxygens from the carboxylic groups of the E125 side chains and the oxygen atoms of the ATP γ-phosphates. Unlike what we had observed in the wild-type structure with $Mg^{2+}$, the R16 side-chains are not in position to coordinate the divalent cation. The coordination distances are consistent with a calcium ion but the coordination number (four ligands) is unusual and there could be additional interactions, with water molecules for example, which we are not currently able to model due to the limited resolution.

These functional and structural experiments establish that the intra-dimer interface site binds both $Mg^{2+}$ and $Ca^{2+}$, further supporting the idea that divalent cations stabilize the square conformation of the RCK domain ring and activate the KtrAB.

## Monovalent cation effect

While establishing the flux assay in choline chloride we tested an assay solution containing 150 mM choline chloride and 1.5 mM KCl. We were surprised to observe that fluorescence quenching was comparable to the sorbitol/$Mg^{2+}$ condition above, with steady-state values of ~80% and a rate constant of 0.062 ± 0.007 $s^{-1}$ (*Figure 8a and c* and *Figure 8—source data 1*).

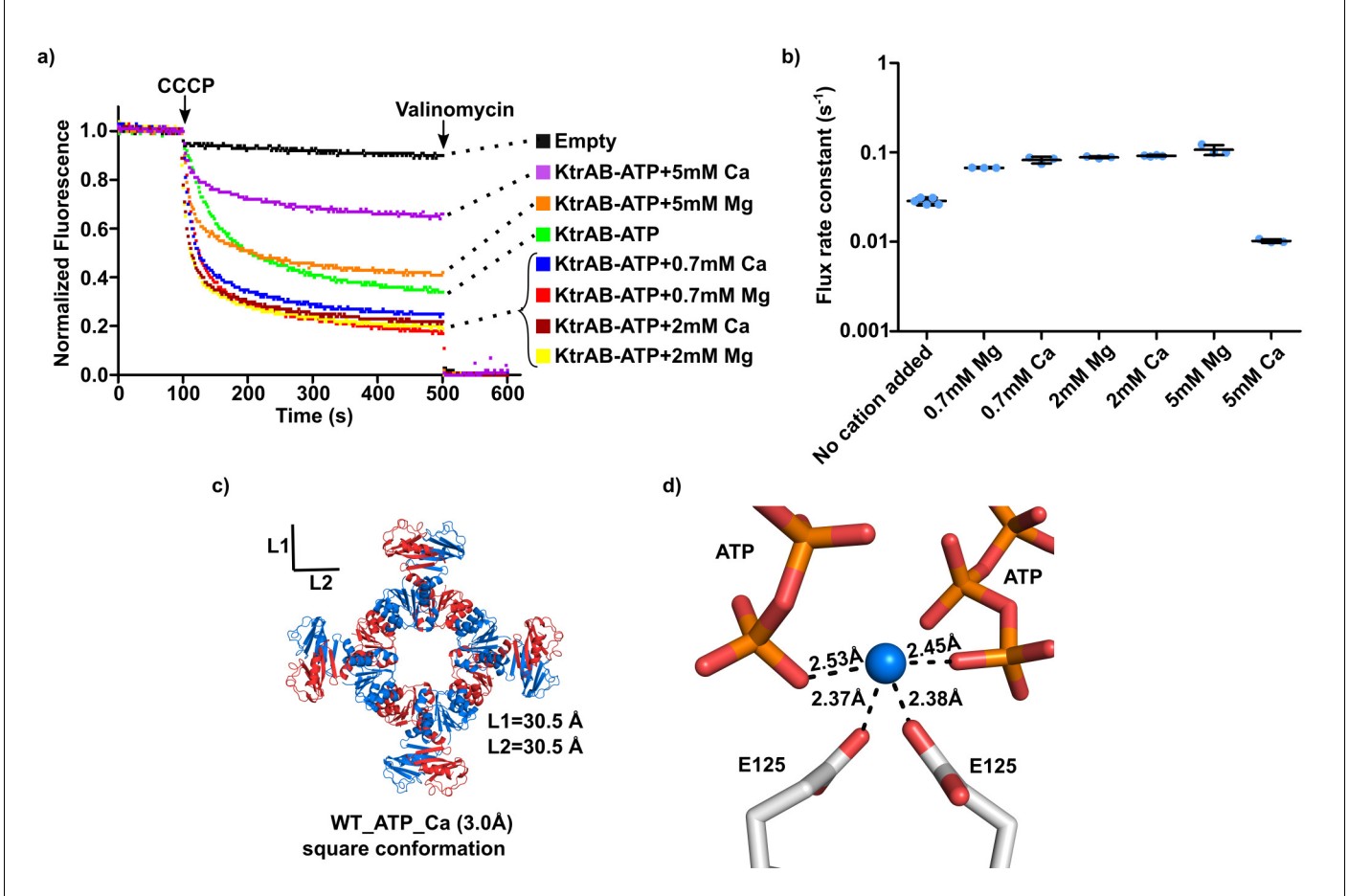

**Figure 7.** Impact of calcium ion on KtrAB activity. (**a**) Proteoliposome flux assay with KtrAB-ATP in an external solution with 20 mM choline chloride and sorbitol. The panel shows representative fluorescence quenching curves. Empty liposomes were used as control. (**b**) Plot of flux rate constants for conditions shown in a). Mean ± SD as well as individual flux rate constants (blue dots) are shown with n = 3–5 of the same proteoliposome preparation. For KtrAB-ATP + 5 mM Mg$^{2+}$ the amplitude of the fast component in the double-exponential fit is slightly lower than of the slow component. Nevertheless, we plotted the rate constant of the fast component as it represents the initial decay phase of the curve, allowing comparison with other rate constants (see *Supplementary file 5*). (**c**) Structure of wild-type KtrA octameric ring with bound ATP supplemented with 5 mM CaCl$_2$. Subunits of the octameric ring are colored alternately blue and red. L1 and L2 correspond to the distances between the Cα of N38 residues located in opposite subunits across the ring. **d**) Calcium-binding site showing cation coordination. Calcium ion is shown as blue sphere. Residues and nucleotides coordinating calcium are shown as stick with coordination distances indicated. See also *Figure 7—source data 1*.

The online version of this article includes the following source data for figure 7:

**Source data 1.** Flux rate constants shown in *Figure 7b*.

Addition of Mg$^{2+}$ had little or no effect in the rate constant (0.058 ± 0.003 or 0.079 ± 0.010 s$^{-1}$ with addition of 0.7 mM or 2 mM Mg$^{2+}$, respectively), confirming that KtrAB is fully active in the presence of 150 mM choline chloride (*Figure 8a and c* and *Figure 8—source data 1*). Lowering the concentration of choline to 20 mM decreased the flux rate by ~2 fold and, as shown above, addition of 0.7 or 2 mM Mg$^{2+}$ reestablishes maximum activity (*Figure 7a and c*).

The dependence of flux on the concentration of choline chloride raised the possibility that the choline cation and/or the chloride anion are also activators of KtrAB. To explore the role of Cl$^-$ we performed the functional assay with 150 mM choline acetate (*Figure 8—figure supplement 1*) and concluded that KtrAB is active in the absence of Cl$^-$ as the rate constants in choline acetate (0.043 ± 0.002 s$^{-1}$) and choline chloride are similar.

To explore the possibility that choline is an activator we analyzed the impact of the E125Q mutation and of ADP in the presence of 150 mM choline. Both KtrA$_{E125Q}$B-ATP and KtrAB-ADP display low ion flux, with rates that are 12- and 16-fold slower than for KtrAB-ATP (*Figure 8b and c* and

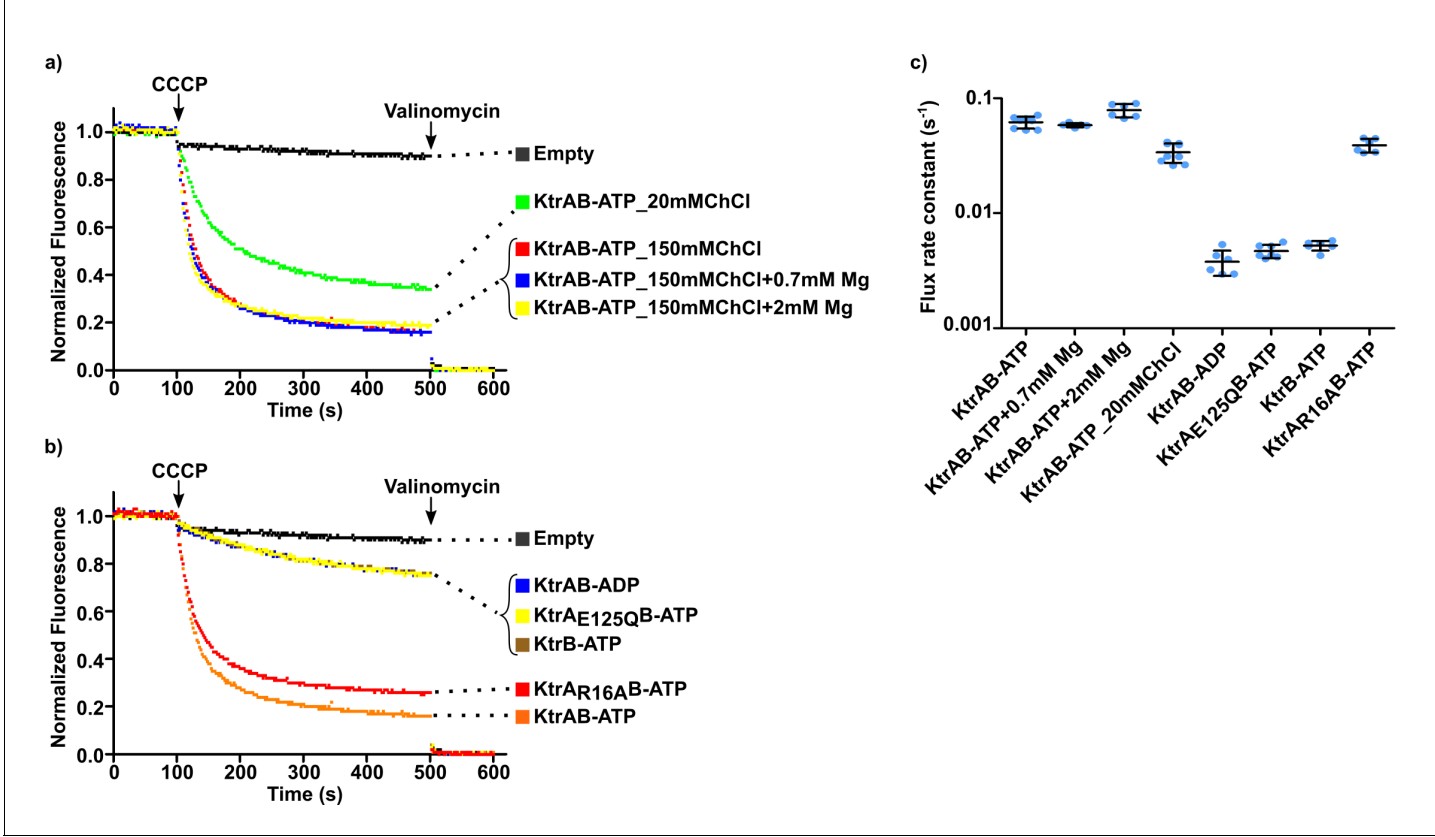

**Figure 8.** Impact of choline on KtrAB flux. (a and b) Proteoliposome flux assay performed with KtrB and wild type and mutant KtrAB with ATP or ADP with choline chloride (ChCl) in the external solution. Panels show representative fluorescence quenching curves. Empty liposomes were used as control. (c) Plot of flux rate constants for conditions shown in a) and b). Mean ± SD as well as individual flux rate constants (blue dots) are shown for n = 5–9 from at least two different proteoliposome preparations. Conditions correspond to 150 mM choline chloride external solution except where stated otherwise. See also *Figure 8—figure supplement 1* and *Figure 8—source data 1*.

The online version of this article includes the following source data and figure supplement(s) for figure 8:

**Source data 1.** Flux rate constants shown in *Figure 8c*.

**Figure supplement 1.** Impact of chloride ion on KtrAB activity.

*Figure 8—source data 1*). Flux observed in these conditions is similar to that of liposomes reconstituted with KtrB alone (*Figure 8b and c*). In contrast, KtrA$_{R16A}$B-ATP is also active in the choline condition without added Mg$^{2+}$ (*Figure 8b and c*). Importantly, using flame ionization spectroscopy we determined that the contaminating levels of Mg$^{2+}$ in the choline solution are undetectable (less than 2 µM), which as shown further down is well below the activating concentrations for this divalent cation. Overall, these data suggest that choline is able to activate KtrAB and that the effect is exerted through the RCK domain ring, involving the same intra-dimer interface site identified for Mg$^{2+}$.

If choline is able to activate KtrAB-ATP, we asked whether the same happens with inorganic monovalent cations. We explored this possibility with Li$^+$ added to the outside solution, as it is likely that KtrAB is more selective for K$^+$ than for Li$^+$ (as demonstrated for the related cation channel TrkHA and for KtrB from *V. alginolyticus* [*Cao et al., 2013*; *Mikušević et al., 2019*]), allowing us to observe flux. In the presence of 5 and 20 mM Li$^+$, flux is higher than in sorbitol alone (~2 and~5 fold, respectively) (*Figure 9* and *Figure 9—source data 1*, confront (cf.) with *Figure 6c*). Addition of Mg$^{2+}$ to either one of these conditions results in a 3- and 1.5-fold increase in the rate constants (0.069 ± 0.001 and 0.070 ± 0.002 s$^{-1}$, respectively), similar to the maximum observed above (cf. *Figure 6c*). Importantly, KtrA$_{E125Q}$B-ATP or KtrB-ATP in the presence of 20 mM Li$^+$ display rates that are ~12 and~15 fold lower than KtrAB-ATP in the same condition, suggesting that the Li$^+$ effect is mediated by the RCK domain (*Figure 9*). In 150 mM Li$^+$ there is a clear decrease in flux rate, down

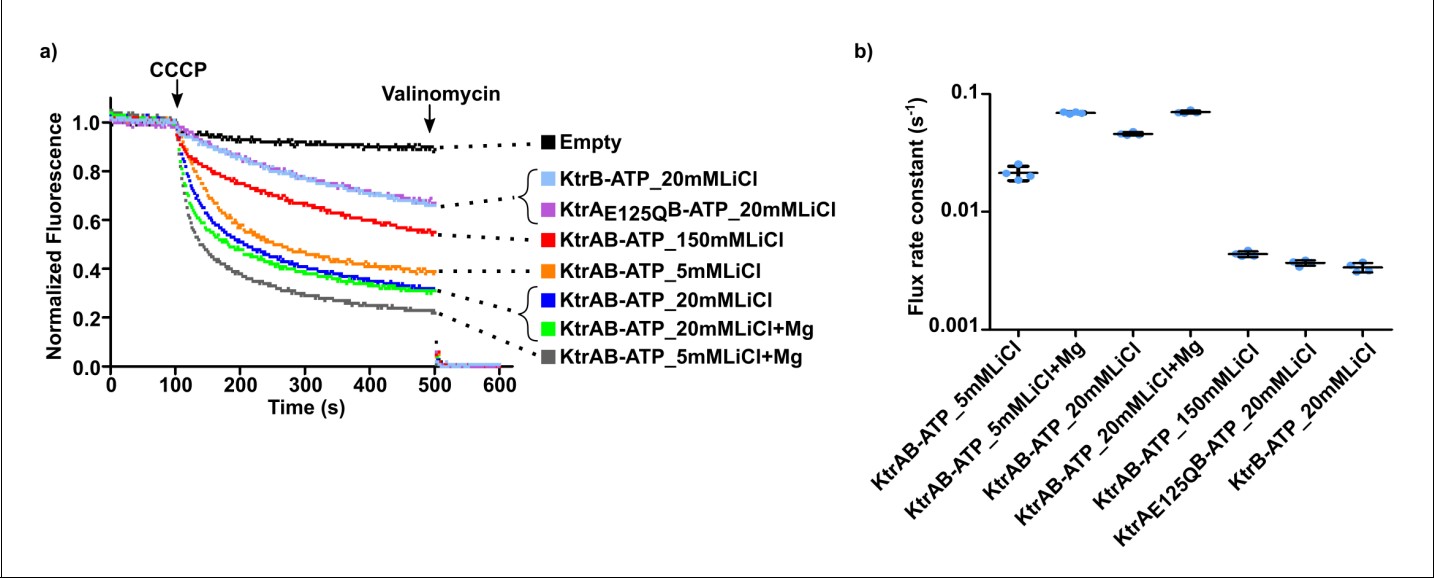

**Figure 9.** Impact of Li$^+$ on KtrAB flux. (**a**) Proteoliposome flux assay performed with KtrB and wild-type and mutant KtrAB with ATP with lithium chloride (LiCl) in the external solution. Osmolarity differences were partially compensated with inclusion of sorbitol in the solution. Panels shows representative fluorescence quenching curves. Empty liposomes were used as control. (**b**) Plot of flux rate constants for conditions shown in a). Mean ± SD as well as individual flux rate constants (blue dots) are shown for n = 4 from one proteoliposome preparation. For conditions in 20 mM Li$^+$ (with and without Mg$^{2+}$) and in 5 mM (Li$^+$ (without Mg$^{2+}$) the amplitude of the fast component of the double-phase exponential fit is either equal or smaller than of the slow component. However, we plot the 'fast' rate constant as it represents the initial decay phase of the curve, allowing comparison with other rate constants (see **Supplementary file 5**). See also **Figure 9—source data 1**.

The online version of this article includes the following source data for figure 9:

**Source data 1.** Flux rate constants shown in **Figure 9b**.

to values that are similar to KtrB-ATP, most likely because in this concentration range Li$^+$, as a permeant ion, affects the build-up of the H$^+$-gradient in the assay (**Figure 9**).

Overall, our data suggest that monovalent cations (both Li$^+$ and choline) are able to activate KtrAB through a mechanism that involves the intra-dimer interface of the RCK domain.

To compare the impact of choline and of Mg$^{2+}$ on the activation mechanism of KtrAB we performed titrations of these two cations using the flux assay (**Figure 10** and **Figure 10—figure supplement 1**). Although osmolarity differences did not affect flux (**Figure 10—figure supplement 2**), we matched the osmolarity of the conditions in the two titration experiments to directly compare results. In addition, we verified that the free concentration of Mg$^{2+}$ in the assay solutions is similar to the added concentration. For this, we spun-down proteoliposomes after running the assay and measured Mg$^{2+}$ in the supernatant using flame ionization spectroscopy. For example, for conditions with 25, 107 and 1156 µM added Mg$^{2+}$, the determined free concentrations were 34 ± 0, 126 ± 8 and 1210 ± 9 µM, respectively. The titrations show that increasing concentrations of both Mg$^{2+}$ and choline result in increased flux rate (**Figure 10a and c**). Fitting the titration curves with a Hill equation yielded a K$_{1/2}$ of activation for Mg$^{2+}$ of 155 ± 17 µM (Hill coefficient 1.8 ± 0.0) and for choline, 25 ± 2 mM for choline with a Hill coefficient 1.9 ± 0.2 (**Figure 10b and d**).

Overall, these data strongly suggest that monovalent cations, including an organic monovalent, are able to activate the KtrAB complex and that this effect is ATP dependent and dependent on an intact intra-dimer interface binding site. Nevertheless, we clearly show that the KtrAB activation mechanism is ~160 fold more sensitive to Mg$^{2+}$ than choline.

## Conserved divalent cation binding site in other nucleotide-dependent RCK domains

Nucleotide-dependent RCK domains are associated with different families of bacterial K$^+$ or cation transporters and we wondered if the intra-dimer interface divalent cation-binding site is conserved

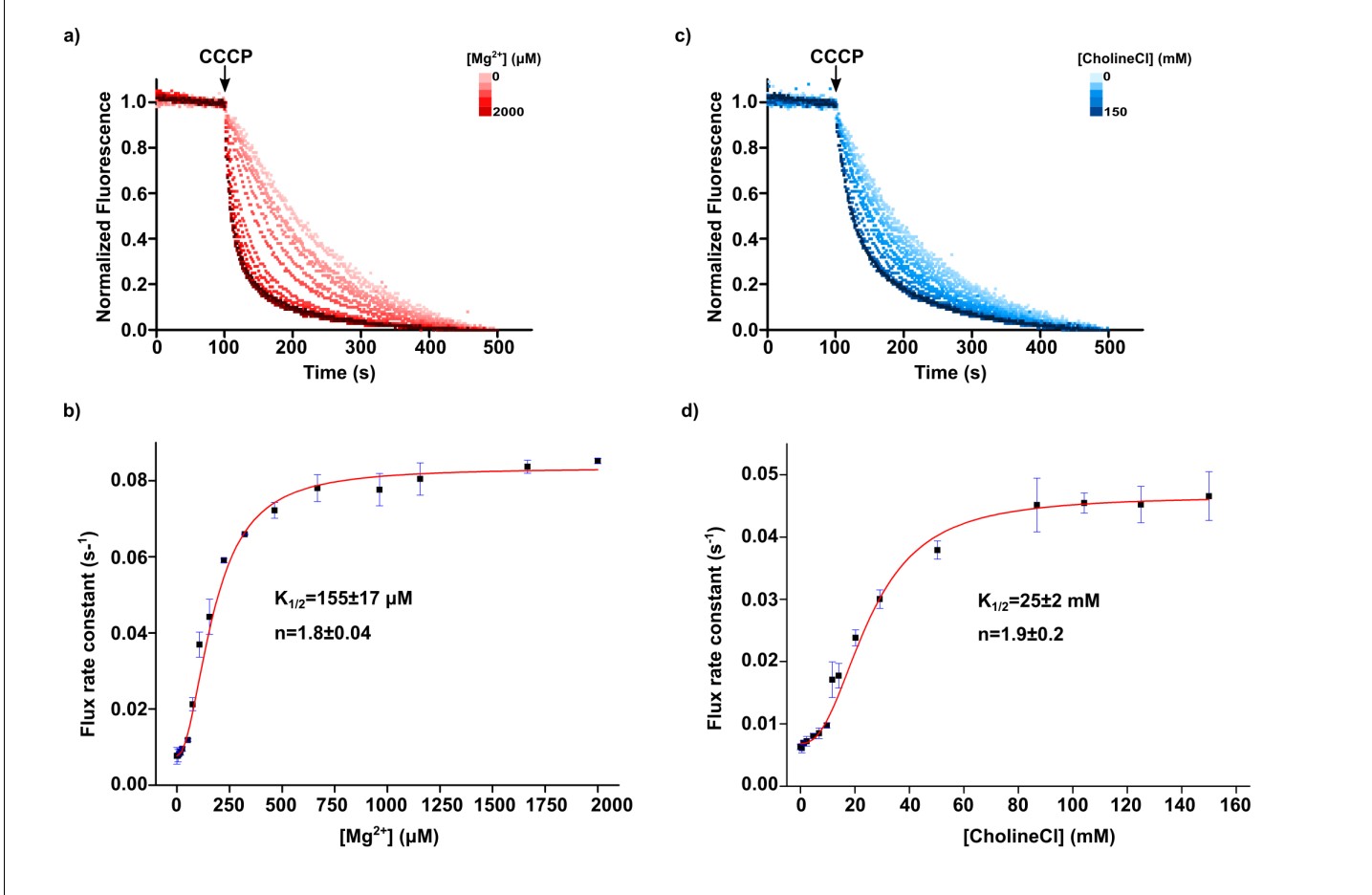

**Figure 10.** $Mg^{2+}$ and Choline titrations. (**a**) Magnesium ion titration of KtrAB-ATP using a proteoliposome flux assay performed with increasing amounts of $MgCl_2$ added to a 300 mM sorbitol external solution. Panel shows representative fluorescence quenching titration curves normalized to their steady-state. Magnesium ion concentration increases with red color intensity from 0 to 2000 µM (**b**) Plot of flux rate constants (mean ± SD, from three separate titrations) as a function of magnesium ion concentration. (**c**) Choline titration of KtrAB-ATP using a proteoliposome flux assay performed with increasing amounts of choline chloride in the external solution while compensating the change in osmolarity with sorbitol. Panel shows representative fluorescence quenching titration curves normalized to their steady-state. Choline chloride concentration increases with blue color intensity from 0 to 150 mM (**d**) Plot of flux rate constants (mean ± SD, from three separate titrations) as a function of choline concentration. Titrations were fitted with a Hill equation, $y = Start + (End - Start) \times \frac{x^n}{K1/2^n + x^n}$ . Mean ± SD of $K_{1/2}$ and of Hill coefficients (n) determined from fits to separate titrations are shown. EDTA was not added to the external solution in these assays. See also *Figure 10—figure supplement 1* and *Figure 10—figure supplement 2*.

The online version of this article includes the following figure supplement(s) for figure 10:

**Figure supplement 1.** Fluorescence raw and normalized data from $Mg^{2+}$ titration.

**Figure supplement 2.** Effect of osmolarity in KtrAB flux.

in those proteins. More specifically, we verified the conservation of a glutamate (equivalent to E125) in the intra-dimer interface. For this we analyzed structures of RCK domains which are known to be nucleotide-dependent by having a bound nucleotide like KtrA or having the classical nucleotide binding site motif associated with a Rossmann fold (GxGxxG[17-18]xE/D; where G is glycine, E glutamate, D aspartate and x any amino acid) (*Bellamacina, 1996*). In this analysis, we did not consider the RCK domain from GsuK since only one of the domains in the RCK dimeric unit binds nucleotides (*Kong et al., 2012*).

First, we analyzed the structure of the KtrA protein from the thermophilic archaean *Methanocal-dococcus jannaschii* (PDB:1LSS) (*Figure 11a*). The structure clearly shows a glutamate residue in the same position as E125 in KtrA from *B. subtilis*. We generated a sequence alignment of KtrA ortho-logs by randomly selecting 2–3 species from all phyla of bacteria and archaea that have Ktr cation

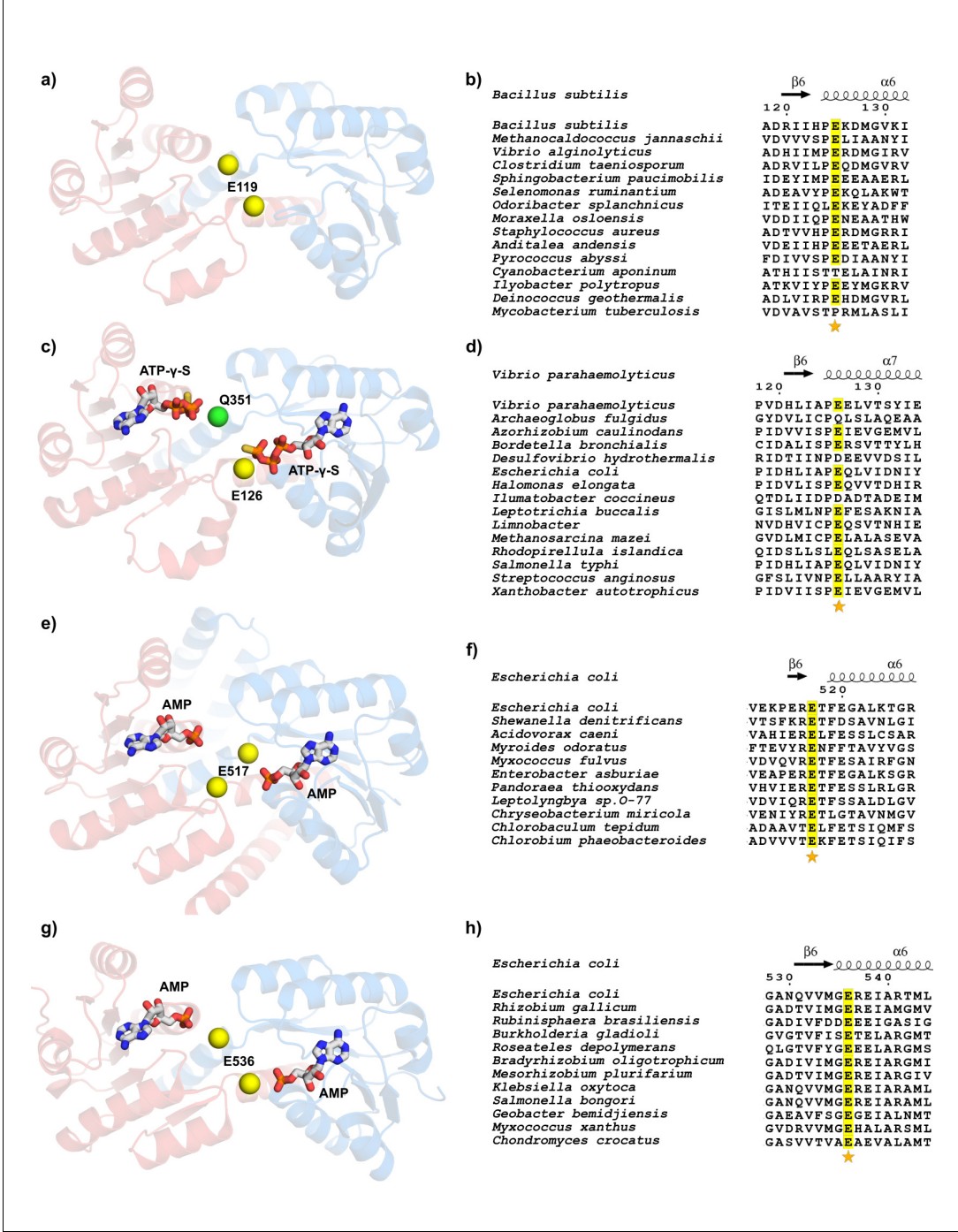

**Figure 11.** Conservation of glutamate residue in the intra-dimer interface of nucleotide-dependent RCK domains. (**a**) View of nucleotide binding sites in *M.jannaschii* KtrA dimer (PDB:1LSS). (**b**) Partial view of sequence alignment of KtrA orthologs with conserved glutamate in yellow. Numbering and secondary structure elements are from *B. subtilis* KtrA. (**c**) View of nucleotide binding sites in *V. parahaemolyticus* TrkA pseudo-dimer (PDB:4J9V). The RCK2 glutamine in an equivalent position to KtrA E125 is shown as a green sphere. (**d**) Partial view of sequence alignment of TrkA orthologs with conserved glutamate in RCK1 in yellow. Numbering and secondary structure elements are from the *V.parahaemolyticus* TrkA. (**e**) View of nucleotide binding sites in *E.coli* KefC C-terminal dimer (PDB:3L9W). (**f**) Partial view of sequence alignment of Kef orthologs with conserved glutamate in yellow. Numbering and secondary structure elements are from the *E.coli* KefC C-terminal. (**g**) View of nucleotide binding sites in *E.coli* YbaL C-terminal dimer (PDB:3FWZ). (**h**) Partial view of sequence alignment of YbaL orthologs with conserved glutamate in yellow. Numbering and secondary structure elements are from the *E.coli* YbaL C-terminal. *Figure 11 continued on next page*

*Figure 11 continued*

Dimer subunits are colored red and blue and conserved glutamates are shown as yellow spheres. Sequence alignments were obtained with MEGA software (*Tamura et al., 2013*) and figures made using ESPript (*Robert and Gouet, 2014*). In all alignments, the position corresponding to KtrA E125 or equivalent is marked with a star. See also *Figure 11—figure supplement 1*, *Figure 11—figure supplement 2*, *Figure 11—figure supplement 3*, *Figure 11—figure supplement 4*, *Figure 11—figure supplement 5*, *Figure 11—figure supplement 6* and *Figure 11—figure supplement 7*.

The online version of this article includes the following figure supplement(s) for figure 11:

**Figure supplement 1.** Sequence alignment of KtrA orthologs.
**Figure supplement 2.** Sequence alignment of RCK1 domains from TrkA orthologs.
**Figure supplement 3.** Sequence alignment of RCK2 domains from TrkA orthologs.
**Figure supplement 4.** Phylogenetic tree of Kef-related proteins.
**Figure supplement 5.** Sequence alignment of C-terminal domains from KefC orthologs.
**Figure supplement 6.** Sequence alignment of C-terminal domains from KefC-like orthologs.
**Figure supplement 7.** Sequence alignment of C-terminal domains from YbaL orthologs.

channels. We cured the alignment to retain only proteins that show the nucleotide binding site sequence motif (*Figure 11b* and *Figure 11—figure supplement 1*). The final alignment has 15 sequences, including KtrA from *B. subtilis*, of which 13 have a glutamate coincident with E125, corresponding to 87% sequence identity. Interestingly, a neighboring proline is also highly conserved, forming the motif PE.

We also inspected the TrkA protein from *Vibrio parahaemolyticus* (PDB:4J9V); this protein assembles with the membrane protein TrkH to form the non-selective TrkHA cation channel (*Figure 11c*). TrkA has two RCK domains in tandem (RCK1 and RCK2), forming a pseudo-dimeric unit, and both domains bind nucleotide. In the intra-dimer interface there is a glutamate in RCK1 in a position similar to that of E125 in KtrA. In contrast, in RCK2 there is a glutamine. A sequence alignment of TrkA orthologs generated using the approach described above shows that the PE sequence motif noted in KtrA is highly conserved in RCK1 (80% sequence identity) (*Figure 11d* and *Figure 11—figure supplement 2*). However, there are no glutamates or aspartates in the E125 position of RCK2 in any of the orthologs (*Figure 11—figure supplement 3*); instead, arginine and lysine residues are common, strongly suggesting that in TrkA the activation mechanism is not dependent on a divalent cation cofactor.

We also found the structure of a RCK octameric ring formed by two different RCK proteins (TM1088A/TM1088B) (*Deller et al., 2015*). Each protein assembles as a homo-dimeric unit and the ring includes two pairs of each dimeric unit, with identical pairs positioned across each other in the ring. As in TrkA, the RCK pair that binds nucleotide does not have the conserved divalent cation site, showing a PA sequence motif, instead of PE, in the intra-dimer interface.

Another nucleotide-dependent RCK family for which there are available structures is that of the regulatory domain of the Kef $K^+/H^+$ antiporters (*Munro et al., 1991*; *Pliotas et al., 2017*; *Roosild et al., 2010*; *Roosild et al., 2009*). Unlike the Ktr and Trk channels where the RCK domain is a separate protein, in Kef antiporters the membrane protein and the RCK domain are in the same polypeptide. In addition, their C-terminal regions lack the C lobe observed in the RCK domains of BK, MthK and KtrA. Despite these differences, the structure of the C-terminal domain of KefC from *E. coli* shows the homo-dimeric unit arrangement and an AMP molecule bound in a similar binding site to KtrA (*Figure 11e*). Strikingly, the KefC structure shows a glutamate at the intra-dimer interface, shifted one residue in the sequence relative to KtrA. However, this is a position where the glutamate can still potentially coordinate a divalent cation, as E125 in KtrA. While analyzing sequences of KefC proteins it became apparent that there are three groups of Kef-related proteins: the KefC proteins (with at least 580 amino acids), the KefC-like group (less than 580 amino acids long) and the YbaL orthologs (*Figure 11—figure supplement 4*). A sequence alignment of the C termini of 11 KefC proteins shows that the glutamate at the intra-dimer interface is 100% conserved (*Figure 11f* and *Figure 11—figure supplement 5*). The sequence motif is RE or TE, and not PE as seen in KtrA. In contrast, in the sequence alignment of KefC-like proteins the glutamate is substituted by a conserved tyrosine or phenylalanine (*Figure 11—figure supplement 6*).

Finally, we analyzed the structure of the nucleotide-dependent RCK domain of YbaL (PDB: 3FWZ), a putative *E. coli* cation antiporter. As stated above, YbaL proteins are related to KefC. This structure also shows the presence of a glutamate in the intra-dimer interface in an equivalent position to that of E125 in KtrA (*Figure 11g*). A sequence alignment reveals that the glutamate is strongly conserved (100% conserved among the 12 sequences) but the sequence motif is primarily GE instead of PE (*Figure 11h* and *Figure 11—figure supplement 7*).

This analysis reveals a striking conservation of glutamates in the intra-dimer interface of many homodimeric nucleotide-dependent RCK domains. These glutamates occupy similar positions to E125 in KtrA, strongly suggesting that the cation-binding site is present in many nucleotide-dependent RCK domains and in particular, that divalent cations are essential cofactors for the activation of these RCK proteins.

## Discussion

We have established that the activation mechanism of KtrA, a nucleotide-dependent RCK domain, involves the divalent cations, $Ca^{2+}$ or $Mg^{2+}$, as cofactors. The divalent cation binds to the intra-dimer interface site of a KtrA dimer, coordinated by E125 and the γ-phosphates from two ATP molecules. This results in the tethering of the two ATP molecules, stabilizing the active state conformation of the octameric ring.

Our data also strongly suggest that monovalent cations like $Li^+$ and choline, an organic monovalent cation, are able to activate KtrAB. This effect is dependent on the RCK domain having bound ATP and an intact intra-dimer interface site, as either ADP or mutation of E125 disrupts activation. This suggests that, like $Mg^{2+}$ or $Ca^{2+}$, monovalent cations bind to the intra-dimer interface site. Although not demonstrated here, it is likely that $K^+$ will also activate KtrAB-ATP, providing an explanation for the partial activity of KtrAB-ATP in the absence of $Mg^{2+}$ (*Figure 6a*). In those experiments, the lumen of the liposomes contains 150 mM $K^+$ and therefore the fraction of KtrAB-ATP molecules that are oriented with KtrA facing the lumen will be activated by $K^+$, giving rise to a flux curve that is intermediate between KtrAB-ADP and KtrAB-ATP-$Mg^{2+}$.

The dimensions of the intra-dimer interface binding site as well as the chemical groups that line this site are ideal for divalent cation binding, as reflected in the KtrA structures with $Mg^{2+}$ and $Ca^{2+}$ and the $K_{1/2}$ of activation ~150 μM, corresponding to 160-fold higher sensitivity of the activation mechanism for $Mg^{2+}$ over choline. Nevertheless, it is easy to imagine that several lithium ions will bind in the intra-dimer interface site, compensating the intense negative electrostatics of the site and stabilizing the close positioning of the γ-phosphate groups in the active conformation. It is harder to imagine how choline binds to the same site but the quaternary ammonium group in choline might adopt the flattened configuration seen in the TBA (tetra-butyl ammonium) molecule bound in the pore of the KcsA $K^+$ channel (*Lenaeus et al., 2014*). In this configuration, the group might be able to slide into the intra-dimer crevice in the non-square conformation and stabilize, at least, a partial closure of the site around the cation, giving rise to activation.

What cation functions as the activation cofactor in the cell? We determined that the $K_{1/2}$ for $Mg^{2+}$ activation is ~150 μM. $Mg^{2+}$ is often found associated with ATP molecules and is by far the most abundant divalent cation in the cytosol of bacteria, with a total concentration of ~100 mM (*Moncany and Kellenberger, 1981*) and a free concentration of ~2 mM (*Alatossava et al., 1985*), 13-fold higher than $K_{1/2}$. In contrast, the concentration of free calcium in bacteria has been estimated at 100–300 nM (*Gangola and Rosen, 1987*; *Jones et al., 1999*). It is therefore reasonable to think that in the cell, the active conformation of the RCK domain ring in KtrAB will involve $Mg^{2+}$ and not $Ca^{2+}$. We have not been able to determine the $K_{1/2}$ of activation for $Li^+$ or $K^+$ using the flux assay and determination of their binding affinity for the intra-dimer interface site by ITC is likely to be complicated by all the interactions established between the cation and other protein chemical groups. However, the reported $K^+$ concentration in the cytosol of *B. subtilis* is ~300 mM, rising to 600–700 mM in certain environmental conditions (*Whatmore et al., 1990*). It is therefore likely that the free concentration of the monovalent cation in the cytosol is sufficient to make a contribution to the activation of KtrAB.

We have determined multiple structures of the KtrA octameric ring. A comparison of the ring structures with bound-ATP and adopting the square conformation revealed that they are very similar to each other, with little differences in the Cα N38 L1/L2 distances and low RMSDs for Cα atoms:

0.56 Å for R16K-ATP and 0.83 Å for R16A-ATP, when compared to WT-ATP (PDB:4J90) (*Figure 3—figure supplement 1*). In contrast, we found a large structural variability in the non-square structures, with four different conformational groups based on Cα N38 L1/L2 distances. This indicates that there is a single well defined square or active conformation for KtrA, while the ring in the non-active (or non-square) conformation is most likely more flexible, giving rise to different ring conformations in the crystals of isolated KtrA.

Although our understanding of the conformational changes occurring in the KtrA ring during activation is well established, the same cannot be said of the membrane protein KtrB. The non-active conformation (ADP-bound) seems well represented by the structure of KtrAB-ADP from *V. alginolyticus* (*Diskowski et al., 2017*), where the flexible KtrA ring appears to adapt its conformation to that of the KtrB dimer, favoring the formation of interactions between the two proteins. In particular, the long transmembrane helix M1D2 from the KtrB subunits interacts with the inner surface of the ring. Binding of ATP and of the cation cofactor to the RCK domain stabilizes the rigid square conformation of the octameric ring. This appears to cause the release of M1D2 from its interaction site on KtrA and bending of the C-terminal end of M1D2 into a short helix, parallel to the membrane surface (*Szollosi et al., 2016*). This change must be propagated to the intramembrane loop region of KtrB, opening the pore gate. However, we do not have a good molecular understanding of the activated-state of KtrB since the ion pore of the membrane protein remains closed in the available KtrAB-ATP structure despite the adoption of the square conformation by KtrA (*Vieira-Pires et al., 2013*). We do not know the reason for this 'incomplete' activation but it is possible that KtrAB-ATP does not adopt the fully active conformation in detergent.

It is worthwhile pointing out that we previously used a $^{86}$Rb$^+$-uptake assay to characterize KtrAB (*Szollosi et al., 2016*; *Vieira-Pires et al., 2013*). With that assay, we observed very low levels of activity for KtrAB-ATP, taking ~60 min to reach 20% of valinomycin signal, that were only 2-fold faster than for KtrAB-ADP. Those values contrast with the levels of activity described here, using the ACMA-liposome K$^+$ flux assay. We now understand that the low KtrAB-ATP activity recorded in those early reports is partly due to a lack of cations in the assay solution, so that many KtrAB-ATP molecules were not in an activated state.

Importantly, analysis of structures and amino acid sequences of nucleotide-dependent RCK domains from several different channel and transporter families demonstrated the presence of a strongly conserved glutamate in the intra-dimer interface, at a position that is equivalent to E125 in KtrA. These conserved glutamates are available for interaction with cations, divalent cations in particular, strongly suggesting that the cation-binding site is conserved and that cations are cofactors in the regulatory mechanism of many nucleotide-dependent RCK domains.

# Materials and methods

**Key resources table**

| Reagent type (species) or resource | Designation | Source or reference | Identifiers | Additional information |
|---|---|---|---|---|
| Strain, strain background (*Escherichia coli*) | TK2420 | other | | Δ(kdpFAB)5 Δ(trkA-mscL') trkD1 |
| Software, algorithm | XDS | DOI: 10.1107/S0907444909047337 | RRID: SCR_015652 | http://xds.mpimf-heidelberg.mpg.de |
| Software, algorithm | Aimless | DOI: 10.1107/S0907444910045749 | RRID: SCR_015747 | http://www.ccp4.ac.uk/html/aimless.html |
| Software, algorithm | Phaser | DOI: 10.1107/S0021889807021206 | RRID: SCR_014219 | https://www.phenix-online.org/documentation/reference/phaser.html |
| Software, algorithm | Coot | DOI: 10.1107/S0907444910007493 | RRID: SCR_014222 | https://www2.mrc-lmb.cam.ac.uk/personal/pemsley/coot/ |
| Software, algorithm | Phenix | DOI: 10.1107/S0907444909052925 | RRID: SCR_014224 | http://phenix-online.org/ |

*Continued on next page*

*Continued*

| Reagent type (species) or resource | Designation | Source or reference | Identifiers | Additional information |
|---|---|---|---|---|
| Software, algorithm | Pymol | PyMOL Molecular Graphics System, Schrödinger, LLC | RRID: SCR_000305 | http://www.pymol.org/ |
| Software, algorithm | GraphPad Prism | GraphPad Software | RRID: SCR_015807 | https://www.graphpad.com |
| Software, algorithm | Origin | OriginLab | RRID: SCR_014212 | https://www.originlab.com/index.aspx?go=PRODUCTS/Origin |
| Software, algorithm | MEGA software | DOI: 10.1093/molbev/mst197 | RRID: SCR_000667 | https://www.megasoftware.net |
| Software, algorithm | ESPript | DOI: 10.1093/nar/gku316 | RRID: SCR_006587 | http://espript.ibcp.fr/ESPript/ESPript/ |
| Chemical compound, drug | ACMA | Sigma-Aldrich | Cat#: A5806 | |
| Chemical compound, drug | CCCP | Sigma-Aldrich | Cat#: 21855 | |
| Chemical compound, drug | Valinomycin | Sigma-Aldrich | Cat#: V0627 | |
| Chemical compound, drug | Choline Chloride | Sigma-Aldrich | Cat#: C7017 | |
| Chemical compound, drug | Sorbitol | Sigma-Aldrich | Cat#: S1876 | |
| Chemical compound, drug | N-Methyl-D-glucamine (NMG) | Sigma-Aldrich | Cat#: 66930 | |
| Chemical compound, drug | *E. coli* Polar Lipid Extract | Avanti Polar Lipids | Cat#: 100600C CAS Number: 1240502-50-4 | |
| Chemical compound, drug | Bio-Beads SM-2 Adsorbents | Bio-Rad | Cat#: 1523920 | |

## Protein expression and purification

Tag-less wild-type KtrA and respective mutants, cloned in a modified pET-24d vector, were over-expressed in *Escherichia coli* BL21(DE3) strain. Cells were grown in LB media at 20°C for 14–16 hr after induction with IPTG. Cell lysis was done in Buffer A (50 mM Tris-HCl pH 7.5, 50 mM KCl, 5 mM DTT, 1 mM EDTA) supplemented with protease inhibitors. Spin-cleared lysate was loaded into an anion exchange column Hi-Trap Sepharose Q-HP (GE Healthcare). KtrA was eluted with a KCl gradient, fractions were pooled and incubated with an ATP-agarose resin (Immobilized γ-Aminophenyl-ATP C10-spacer from Jena Bioscience) overnight at 4°C with gentle agitation. Beads were then washed thoroughly with Buffer B (50 mM Tris-HCl pH 8.0, 150 mM KCl, 1 mM TCEP, 1 mM EDTA) and protein was eluted in Buffer B supplemented with 5 mM ADP or ATP (sodium salts). Protein was concentrated to ~5 mg/ml and further purified by size exclusion chromatography with a Superdex-S200 (GE Healthcare) column using Buffer C (50 mM Tris-HCl pH 7.5, 150 mM KCl, 5 mM DTT, 0.2 mM EDTA). KtrA was supplemented with 1 mM ATP or ADP.

N-terminal Strep-tagged KtrB, cloned in a pRSFDuet-1 vector was over-expressed in *Escherichia coli* BL21(DE3) strain. Cells grown in LB media at 37°C were induced with IPTG for 2.5 hr; 1 mM BaCl2 was added together with inducer. Cells were lysed in Buffer D (50 mM Tris-HCl pH 8.0, 120 mM NaCl, 30 mM KCl, 1 mM EDTA) supplemented with protease inhibitors and KtrB was extracted with 40 mM DDM (*n*-dodecyl-β-D-maltoside, Sol-grade from Anatrace) overnight at 4°C with gentle agitation. Spin-cleared lysate was loaded into a Strep-Tactin Sepharose resin (IBA Lifesciences) and washed with Buffer E (50 mM Tris-HCl pH 8.0, 120 mM NaCl, 30 mM KCl, 1 mM DDM, 5 mM DTT, 1 mM EDTA, 1 mM ATP or ADP). Purified KtrA-ATP or ADP was then added to the beads and incubated for 30 min at 4°C to assemble the KtrAB complex. Beads were thoroughly washed with Buffer E and KtrAB was eluted with Buffer E supplemented with 5 mM desthiobiotin. Protein was dialysed overnight at 4°C against Buffer F (10 mM HEPES, 7 mM NMG (N-methylglucamine), pH 8.0, 150 mM

KCl, 0.5 mM DDM, 5 mM DTT, 1 mM EDTA) in the presence of thrombin for tag cleavage. Protein was further purified by size-exclusion chromatography with a Superdex-S200 (GE Healthcare) column using Buffer F containing 0.2 mM EDTA. Eluted fractions were concentrated to 1 mg/ml and used for proteoliposome preparation when required. KtrB alone was purified in the same manner as the KtrAB complexes without the KtrA addition step. No nucleotide was added during dialysis and size-exclusion chromatography.

## Crystallization

KtrA-ATP and KtrA-ADP (WT and mutants) were purified in buffer C and supplemented with 1 mM ATP or ADP, respectively. Crystals were grown using the sitting-drop vapor diffusion method at 20° C.

KtrA$_{WT}$-ATP-Ca$^{2+}$ crystals were obtained by supplementing KtrA-ATP with 5 mM CaCl$_2$ after size-exclusion chromatography. Crystals grew in 100 mM HEPES-NaOH pH 8.0, 5% PEG 6000, 2.5% MPD (2-Methyl-2,4-pentanediol). Before being flash-cooled in liquid nitrogen, crystals were cryoprotected by sequential transfer to 100 mM HEPES pH 8.0, 6% PEG 6000, 2.5% MPD, 0.5 mM ATP, 2.5 mM CaCl$_2$, supplemented with 15% and 30% glycerol.

KtrA$_{R16K}$-ATP crystals grew in 100 mM Tris-HCl pH 8.5, 8% PEG 8000. Before being flash-cooled in liquid nitrogen, crystals were cryoprotected using 20% glycerol by adding 100 mM Tris-HCl pH 8.5, 10% PEG 8000, 0.5 mM ATP, 25% glycerol solution directly to the drop and equilibrating before freezing.

KtrA$_{R16K}$-ADP crystals grew in 100 mM HEPES-NaOH pH 7.5, 6 to 8% PEG 8000, 20% ethylene glycol and were directly flash-cooled in liquid nitrogen.

KtrA$_{R16A}$-ATP crystals grew in 100 mM HEPES-NaOH pH 7.5, 2% PEG 6000, 1% MPD. Before being flash-cooled in liquid nitrogen, crystals were cryoprotected with 100 mM HEPES-NaOH pH 7.5, 6% PEG 6000, 25% ethylene glycol solution.

KtrA$_{R16A}$-ADP crystals grew in a solution optimized from solution number 2–30 of the Morpheus crystallization kit (Molecular Dimensions, containing 5% PEG 8000% and 20% ethylene glycol. Before being flash-cooled in liquid nitrogen, crystals were cryoprotected by transferring to mother liquor containing 12% PEG 8000.

KtrA$_{E125Q}$-ATP crystals grew in 100 mM MES pH 6.5, 6% PEG 4000, 20% glycerol. Before being flash-cooled in liquid nitrogen, crystals were cryoprotected with 100 mM MES pH 6.5, 12% PEG 4000, 20% glycerol solution.

KtrA$_{E125Q}$-ADP crystals grew in 100 mM ADA pH 6.5, 4% PEG 4000, 20% glycerol. Before being flash-cooled in liquid nitrogen, crystals were cryoprotected by transferring to 100 mM ADA pH 6.5, 12% PEG 4000, 20% glycerol solution.

KtrA$_{A80P}$-ATP crystals grew in 100 mM HEPES-NaOH pH 7.5, 1% PEG 6000, 2% MPD. Before being flash-cooled in liquid nitrogen, crystals were cryoprotected with 100 mM HEPES-NaOH pH 7.5, 6% PEG 6000, 2% MPD, 25% ethylene glycol solution.

KtrA$_{A80P}$-ADP crystals grew in 100 mM HEPES-NaOH pH 7.5, 2% PEG 4000, 20% glycerol. Crystals were cryoprotected with to 100 mM HEPES-NaOH pH 7.5, 12% PEG 4000, 20% glycerol solution, before being flash-cooled in liquid nitogen.

## Data processing and structure determination

Diffraction data were collected at ALBA Synchrotron, Barcelona, Spain; European Synchrotron Radiation Facility (ESRF), Grenoble, France, and French National Synchrotron Source (SOLEIL), Gif-sur-Yvette, France. Data were processed using XDS (*Kabsch, 2010*) and AIMLESS (CCP4 program suite) (*Winn et al., 2011*).

KtrA$_{R16K}$-ATP and KtrA$_{WT}$-ATP-Ca$^{2+}$ structures were determined by molecular replacement with Phaser (*McCoy et al., 2007*) using a KtrA$_{WT}$-ATP dimer (PDB: 4j90) and a KtrA$_{R16K}$-ATP dimer (PDB:6S2J) as search models, respectively. Final models were obtained after several manual model building rounds in Coot (*Emsley et al., 2010*) and refinement with Phenix (*Adams et al., 2010*).

The lower resolution structures were solved by molecular replacement with Phaser (*McCoy et al., 2007*) using a KtrA$_{WT}$-ATP dimer as template (PDB: 4j90). Refinement was limited to an initial rigid body and TLS single group refinement in Phenix (*Adams et al., 2010*), followed by minimal

adjustments to resolve stereo-chemical violations in Coot and refinement with secondary structure and non-crystallographic restraints in Phenix.

## Complementation assay

The complementation assay with the *E. coli* TK2420 strain was performed as previously described (*Albright et al., 2007*; *Vieira-Pires et al., 2013*) with modifications. Wild-type KtrB and KtrA (wild-type or mutants) were cloned into a dicistronic constitutive expression plasmid (pKCe). Empty pKCe plasmid was used as negative control. Transformed cells were plated on LBK (Luria–Bertani broth where NaCl is replaced by KCl) agar plates. Three individual colonies for each protein version were picked and grown separately in 5 mL of minimal media containing 30 mM K$^+$, overnight at 37°C. A 5 µl aliquot of the overnight cultures was then used to inoculate 5 ml of minimal media prepared with different K$^+$ concentrations (0.1, 0.3, 1, 2, 6, 10, 30 or 115 mM) and incubated at 37°C. Optical density at 595 nm was measured after 16 hr.

## Preparation of proteoliposomes

Preparation follows previously described methods with some modifications (*Szollosi et al., 2016*; *Vieira-Pires et al., 2013*). Briefly, *E. coli* polar lipids (Avanti) were resuspended at 10 mg/mL in Swelling Buffer (150 mM KCl, 10 mM HEPES, 7 mM NMG, pH 8.0, 0.2 mM EDTA) and solubilized by adding 40 mM DM (n-Decyl-β-D-Maltoside) (Sol-grade from Anatrace). KtrB alone and KtrAB complexes were purified just before reconstitution and supplemented with ATP or ADP to reach a final concentration of 0.1 mM after reconstitution. Protein was added to solubilized lipids at 1:100 (w:w) protein-to-lipid ratio and incubated for 30 min at room temperature. In control liposomes (empty), Swelling Buffer was added to the lipids instead of protein solution. Detergent was removed using adsorbing SM-2 Biobeads (Biorad): the protein-lipid mix was incubated twice with fresh Biobeads at 10:1 (w:w) bead-to-detergent ratio at room temperature for 1 hr and then incubated overnight at 4°C at a 20:1 (w:w) bead-to-detergent ratio. Proteoliposomes were then frozen in liquid nitrogen and stored at −80°C.

## Fluorescence-based K$^+$ flux assay

The assay was adapted from a previously described method (*Su et al., 2016*). Proteoliposomes were thawed in a 37°C water bath for 15 min, briefly sonicated and kept at room temperature. In assays with the sorbitol external solution, proteoliposomes were diluted 100-fold in Flux Buffer (10 mM HEPES and 7 mM NMG (pH 8.0), 150 mM sorbitol, 0.1 mM ATP; 0.2 mM EDTA) to establish the K$^+$ gradient. The pH-sensitive dye 9-amino-6-chloro-2-methoxyacridine (ACMA) was then added to the proteoliposomes to a final concentration of 550 nM, from a 111 µM stock in DMSO. Fluorescence was monitored every 2 s ($\lambda_{Ex}$=410 nm, $\lambda_{Em}$=480 nm), using a 104F-QS 10 mm quartz cuvette with a small magnet in a Horiba FluoroMax four spectrometer. After measuring the initial baseline fluorescence during 100 s, the assay was initiated by adding 1 µM of the H$^+$ ionophore carbonyl cyanide *m*-chlorophenyl hydrazine (CCCP) from a 200 µM stock in HNE Buffer (10 mM HEPES, 7 mM NMG (pH 8.0), 1 mM EDTA). CCCP enables H$^+$ entry into the liposomes to counterbalance the negative potential created upon K$^+$ efflux, leading to ACMA fluorescence signal quenching. The flux signal was monitored for 400 s. For each experimental condition, one additional replica was performed in which, after baseline fluorescence measurement and CCCP addition, 296 nM of the K$^+$ ionophore valinomycin was quickly added from a 60 µM stock in DMSO and flux signal was monitored for an additional 100 s. This extra measurement was performed to avoid valinomycin contamination of the cuvette between sample measurements and was done every day for each proteoliposome preparation and external solution used. Valinomycin incorporates into all liposomes and mediates K$^+$ efflux, indicating the total flux of the liposome population. The maximum value of fluorescence quenching change obtained was used to normalize the data.

For assays with the choline or lithium external solution, the 150 mM sorbitol in the Flux Buffer was partially or totally replaced by choline chloride, choline acetate or lithium chloride.

Fluorescence quenching curves were normalized individually as follows:

$$NF = \frac{F - Fval}{Fini - Fval}$$

NF is the normalized fluorescence, F is the measured fluorescence in arbitrary units, $F_{ini}$ corresponds to the last baseline point measured before CCCP addition and $F_{val}$ corresponds to the lowest point measured after valinomycin addition. For magnesium and choline titrations, normalization was performed taking into account the steady-state of each curve instead of the maximum value of fluorescence quenching change obtained with valinomycin. In these cases, $F_{val}$ was replaced in the previous expression by $F_{ss}$ that corresponds to the average of the last 50 s of the assay (450–500 s).

## Determination of magnesium ion concentration in flux assay solutions

For several of the flux assay mixtures used for the $Mg^{2+}$ titration, we collected 1 mL of the assay mixture after running the assay. Proteoliposomes were removed from these samples by centrifuging 800 µL at 267,000 g for 25 min at 4°C. 700 µL of the resulting supernatant were used for determination of magnesium ion concentration by flame ionization spectroscopy analysis at the FCUP|DQB-Lab and Services (University of Porto). Buffer solutions used in the flux assay were also tested for magnesium contamination.

## Quantification of flux rate constants

Individual fluorescence quenching curves (after CCCP addition) were fitted with a one-phase exponential equation: $y = y0 + A1 \times e^{-(x-x0)/\tau}$ or a two-phase exponential equation $y = y0 + A1 \times e^{-(x-x0)/\tau fast} + A2 \times e^{-(x-x0)/\tau slow}$ using the Origin software, where A1 and A2 are amplitudes, x0 is the initial fit point, y0 corresponds to the plateau and $\tau$ is the time constant.

In each case, the simplest model that explained the data was chosen. The first point measured immediately after CCCP addition was not considered in this fit due to uncertainty in the fluorescence measurement after compound addition.

Flux rate constants were determined by inverting the respective time constants (rate constant = 1/$\tau$).

## Acknowledgements

We thank access to ALBA (XALOC), ESRF (ID23-1) and Soleil (PROXIMA 1 and 2a) synchrotrons and technical support provided by the i3S scientific platform 'Biochemical and Biophysical Technologies' and FCUP|DQB-Lab and Services. Work was supported by *Fundação Luso-Americana para o Desenvolvimento* through the FLAD Life Science 2020 award entitled 'Bacterial $K^+$ transporters are potential antimicrobial targets: mechanisms of transport and regulation' and by FEDER - Fundo Europeu de Desenvolvimento Regional funds through the COMPETE 2020 - Operational Programme for Competitiveness and Internationalization (POCI), Portugal 2020, and by Portuguese funds through FCT - Fundação para a Ciência e a Tecnologia/Ministério da Ciência, Tecnologia e Ensino Superior in the framework of the projects POCI-01–0145-FEDER-029863 (PTDC/BIA-BQM/29863/2017) and 'Institute for Research and Innovation in Health Sciences' (POCI-01–0145-FEDER-007274).' CMT-D was supported by FCT fellowship (SFRH/BD/123761/2016) and FF was supported by FCT fellowship (SFRH/BPD/102753/2014).

## Additional information

### Funding

| Funder | Grant reference number | Author |
|---|---|---|
| Fundação Luso-Americana para o Desenvolvimento | FLAD Life Science 2020 | João H Morais-Cabral |
| Fundação para a Ciência e a Tecnologia | POCI-01-0145-FEDER-029863 (PTDC/BIA-BQM/29863/2017) | João H Morais-Cabral |
| Fundação para a Ciência e a Tecnologia | POCI-01-0145-FEDER-007274 | João H Morais Cabral |
| Fundação para a Ciência e a Tecnologia | SFRH/BD/123761/2016 | Celso M Teixeira-Duarte |

| Fundação para a Ciência e a Tecnologia | SFRH/BPD/102753/2014 | Fátima Fonseca |
| Fundo Europeu de Desenvolvimento Regional | COMPETE 2020 | João H Morais Cabral |

The funders had no role in study design, data collection and interpretation, or the decision to submit the work for publication.

## Author contributions

Celso M Teixeira-Duarte, Conceptualization, Data curation, Formal analysis, Investigation, Methodology, Writing - original draft, Writing - review and editing; Fátima Fonseca, Conceptualization, Data curation, Formal analysis, Supervision, Funding acquisition, Investigation, Methodology, Writing - original draft, Writing - review and editing; João H Morais-Cabral, Conceptualization, Formal analysis, Supervision, Funding acquisition, Project administration

## Author ORCIDs

Celso M Teixeira-Duarte (iD) http://orcid.org/0000-0002-8036-7069
Fátima Fonseca (iD) http://orcid.org/0000-0001-7979-5907
João H Morais-Cabral (iD) https://orcid.org/0000-0002-4461-9716

## Decision letter and Author response

Decision letter https://doi.org/10.7554/eLife.50661.sa1
Author response https://doi.org/10.7554/eLife.50661.sa2

# Additional files

## Supplementary files

• Supplementary file 1. PDB structures of prokaryotic proteins containing RCK domains.

• Supplementary file 2. $C\alpha$ D36-D36 intra-dimer distances in KtrA octameric rings. $^{\$}$ R16A-ADP, E125Q-ATP and E125Q-ADP have two different octameric rings in the crystal, indicated by (1) or (2).

• Supplementary file 3. $C\alpha$ N38 distances (L1/L2) in KtrA octameric rings. *Distances measured on the two faces of the ring are slightly different and we have named the front face as the one with the larger individual distance value. Values mentioned in the main text correspond to front face. $^{\#}$ Non-square conformations were clustered in four groups according to L1/L2. $^{\$}$ R16A-ADP, E125Q-ATP and E125Q-ADP have two different octameric rings in the crystal, indicated by (1) or (2).

• Supplementary file 4. $B$ factors of divalent cation and coordinating atoms in R16K-ATP and WT-ATP KtrA structures after refinement with magnesium or calcium.

• Supplementary file 5. Parameters extracted from fits of fluorescence quenching curves with one-phase ($\tau$) or two-phase exponential equations ($\tau_{fast}$ and $\tau_{slow}$). Values shown are the mean $\pm$ SD values calculated from parameters from individual fits. Rate constants referred to in the main-text and figures were calculated as the mean of the inverse of $\tau$ extracted from individual fits. * For KtrA$_{E125Q}$B-ATP in 150 mM sorbitol the plateau had to be fixed during fitting since the signal is closer to a linear decay.

• Transparent reporting form

## Data availability

Diffraction data have been deposited in PDB under the accession code 6S2J, 6S5B, 6S5D, 6S7R, 6S5N, 6S5O, 6S5E, 6S5G,6S5C. Source data files have been provided for Figures 6, 7, 8 and 9.

The following datasets were generated:

| Author(s) | Year | Dataset title | Dataset URL | Database and Identifier |
|---|---|---|---|---|
| Teixeira-Duarte CM, Fonseca F, Morais-Cabral JH | 2019 | Square conformation of KtrA R16K mutant ring with bound ATP | http://www.rcsb.org/structure/6S2J | RCSB Protein Data Bank, 6S2J |

| Teixeira-Duarte CM, Fonseca F, Morais-Cabral JH | 2019 | Non-square conformation of KtrA R16K mutant ring with bound ADP | http://www.rcsb.org/structure/6S5B | RCSB Protein Data Bank, 6S5B |
| Teixeira-Duarte CM, Fonseca F, Morais-Cabral JH | 2019 | Square conformation of KtrA R16A mutant ring with bound ATP | http://www.rcsb.org/structure/6S5D | RCSB Protein Data Bank, 6S5D |
| Teixeira-Duarte CM, Fonseca F, Morais-Cabral JH | 2019 | Non-square conformations of KtrA R16A mutant rings with bound ADP | http://www.rcsb.org/structure/6S7R | RCSB Protein Data Bank, 6S7R |
| Teixeira-Duarte CM, Fonseca F, Morais-Cabral JH | 2019 | Non-square conformations of KtrA E125Q mutant rings with bound ATP | http://www.rcsb.org/structure/6S5N | RCSB Protein Data Bank, 6S5N |
| Teixeira-Duarte CM, Fonseca F, Morais-Cabral JH | 2019 | Non-square conformations of KtrA E125Q mutant rings with bound ADP | http://www.rcsb.org/structure/6S5O | RCSB Protein Data Bank, 6S5O |
| Teixeira-Duarte CM, Fonseca F, Morais-Cabral JH | 2019 | Non-square conformation of KtrA A80P mutant ring with bound ATP | http://www.rcsb.org/structure/6S5E | RCSB Protein Data Bank, 6S5E |
| Teixeira-Duarte CM, Fonseca F, Morais-Cabral JH | 2019 | Non-square conformation of KtrA A80P mutant ring with bound ADP | http://www.rcsb.org/structure/6S5G | RCSB Protein Data Bank, 6S5G |
| Teixeira-Duarte CM, Fonseca F, Morais-Cabral JH | 2019 | Square conformation of KtrA WT ring with bound ATP and calcium | http://www.rcsb.org/structure/6S5C | RCSB Protein Data Bank, 6S5C |

The following previously published dataset was used:

| Author(s) | Year | Dataset title | Dataset URL | Database and Identifier |
| --- | --- | --- | --- | --- |
| Vieira-Pires RS, Morais-Cabral JH | 2013 | Square-shaped octameric structure of KtrA with ATP bound | http://www.rcsb.org/structure/4J90 | RCSB Protein Data Bank, 4J90 |

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
