## [Decision Letter]

**Acceptance summary:**

This study examines the structural basis of ATP-dependent activation of KtrA, an RCK domain that forms a component of the bacterial KtrAB K-efflux complex. The authors combine X-ray crystallography on octameric RCK domain rings with flux assays on reconstituted KtrAB complexes to determine the structural and functional consequences of ligand binding and/or mutations at the RCK domain interface. Binding of ATP is shown to stabilize a single fourfold symmetric square conformation, whereas ADP binding results in various non-square conformations. Mutation of a conserved arginine (R16) near the ATP γ phosphate distorts the structure of the ADP-bound, but not that of the ATP-bound, form of the ring, and does not impair cation fluxes through KtrAB. In contrast, mutation of a nearby glutamate (E125) prevents formation of the active square form, and eliminates cation fluxes through KtrAB complexes. At the RCK domain interface of ATP-bound rings a divalent cation (Mg^2+^ or Ca^2+^) is identified which bridges the two ATP γ phosphates and is coordinated by the E125 side chains. Cation fluxes through KtrAB complexes are shown to be divalent cation dependent: complete removal of Mg^2+^ and Ca^2+^ reduces cation fluxes by ~20-fold. The conclusion is that divalent cations are important cofactors that stabilize the active, square conformation of nucleotide-dependent RCK domains. This is a conceptually novel finding that significantly increases our understanding of RCK domain function.

**Decision letter after peer review:**

Thank you for submitting your article "Activation of a nucleotide-dependent RCK domain requires binding of a divalent cation cofactor to a conserved site" for consideration by *eLife*. Your article has been reviewed by three peer reviewers, including László Csanády as the Reviewing Editor and Reviewer #1, and the evaluation has been overseen by Richard Aldrich as the Senior Editor. The following individual involved in review of your submission has agreed to reveal their identity: Ming Zhou (Reviewer #2).

The reviewers have discussed the reviews with one another and the Reviewing Editor has drafted this decision to help you prepare a revised submission.

Summary:

This study examines the structural basis of ATP-dependent activation of KtrA, an RCK domain that forms a component of the bacterial KtrAB K-efflux complex. The authors combine X-ray crystallography on octameric RCK domain rings with flux assays on reconstituted KtrAB complexes to determine the structural and functional consequences of ligand binding and/or mutations at the RCK domain interface. Binding of ATP is shown to stabilize a single fourfold symmetric square conformation, whereas ADP binding results in various non-square conformations. Mutation of a conserved arginine (R16) near the ATP γ phosphate distorts the structure of the ADP-bound, but not that of the ATP-bound, form of the ring. In contrast, mutation of a nearby glutamate (E125) prevents formation of the active square form, and eliminates cation fluxes through KtrAB complexes. At the RCK domain interface of ATP-bound rings a Mg^2+^ ion is identified which bridges the two ATP γ phosphates and is coordinated by the E125 side chains. Cation fluxes through KtrAB complexes are shown to be divalent cation dependent: complete removal of Mg^2+^ and Ca^2+^ reduces cation fluxes by ~20-fold. The conclusion is that divalent cations are important cofactors that stabilize the active, square conformation of nucleotide-dependent RCK domains. This is a conceptually novel finding which promotes our understanding of RCK domain function. The general conclusions seem solid, but some concerns were raised by the reviewers, which will need to be addressed.

Essential revisions:

1) We would like to see flux assays performed on the R16A mutant.

First, you showed that E125A does not assume a square conformation in any of the solved structures, and then showed that E125A is also defective in gating in the flux assay. This is all consistent with the presented gating model. It would be good to show that R16A can still gate the channel because it can adopt both square and non-square conformations. That would strengthen the proposed correlation. Second, you went on to show that E125 coordinates a Mg^2+^ ion, and that the Mg^2+^ ion is required for gating. Although R16 was also found to coordinate the same Mg^2+^ ion, the R16A structure shows that both Mg and ATP are still bound. R16A should be functional based on these observations.

2) Mg^2+^ binding affinity should be better quantitated.

The estimation of the apparent affinity of KtrAB for Mg^2+^ is confusing. You conclude that the K_D_ of the channel for Mg^2+^ is ~1 nM, but several arguments seem incompatible with that claim. First, in 5 mM EDTA+6 mM Mg^2+^ (free [Mg^2+^]=~ 1 mM) the initial rate seems submaximal (Figure 7D, purple curve). Second, in your calculations you use a K_D_ of ~2 nM for the Mg-EDTA complex (subsection “Determination of KtrAB binding affinity for Mg^2+^”). However, this is the K_D_ of the completely deprotonated form of Mg-EDTA. The apparent K_D_ is strongly pH dependent, at pH=8.0 it is ~0.3-0.5 uM. Maxchelator or other free software should be used to estimate the apparent K_D_ of Mg-EDTA under your experimental conditions. Third, the origin of the equations used for calculating the apparent affinity of the KtrAB complex for Mg^2+^ is unclear, and needs explanation. Fourth, you claim that the estimate of a K_D_ ~1 nM was obtained assuming 2 mM contaminating Mg^2+^ in the buffer. However, in the presence of 2 mM Mg^2+^ plus 5 mM EDTA (pH=8.0) the free [Mg^2+^] is ~250 nM (Maxchelator), and under those conditions the flux rate is ~half-maximal (Figure 7E).

The manuscript would be greatly improved if you could provide a more reliable estimate of the affinity of KtrAB for Mg^2+^. One straightforward approach would be to measure the total [Mg^2+^] in their 150 mM choline chloride buffer (e.g., using Mg-Green fluorescence). With that information at hand, free [Mg^2+^] could be calculated for each of the tested experimental conditions using Maxchelator. The plot in Figure 7E could then be replaced by a plot which displays initial flux rates as a function of free [Mg^2+^] (rather than EDTA), and fitted to a simple hyperbolic binding equation to obtain the apparent K_D_. Alternatively, ITC could be used to measure divalent cation binding affinity of KtrA in the presence of ATP and ADP.

3) Quantitation of the flux assays needs revision.

In principle, the steady-state fluorescence should be independent of the initial flux rate, and should reflect the fraction of liposomes that lack a reconstituted (active) KtrAB complex. A comment on the possible reasons for the strong correlation between the observed initial flux rates and the final steady-state fluorescence levels (faster initial fluxes tend to be associated with lower steady-state fluorescences) would be appreciated. Given the variable levels of steady-state fluorescence it seems inappropriate to report flux rates as initial slopes, since the initial slope depends not only on the flux rate, but also on the total amplitude of the signal (F_initial_ – F_steady-state_). One way to obtain more reliable flux rates would be to fit the fluorescence time courses with single decaying exponentials and report the fitted rate constants. Alternatively, the "normalized fluorescence" signal could be redefined as NF=(F – F_ss_)/(F_ini_ – F_ss_), where F_ss_ is the steady-state fluorescence (before valinomycin addition). This transformation would equalize the total amplitude of each signal and would make comparison of initial slopes more meaningful.

4) Subsection “Magnesium binding site”: "instead of water molecules, nitrogen atoms of the R16 guanidinium group appear to complete the coordination shell of the divalent cation"

Are there any other known examples of an arginine side chain coordinating a divalent cation? Mg does not have the character of a transition metal, for which one might expect coordination by N atoms, and the N-Mg distances shown in Figure 5D are much longer than what would be expected for direct coordination. Is it possible that there are water molecules present in the coordination shell whose density is weak or unresolved, and that there is uncertainty in the orientation of the R16 guandinium in this 3.2 A structure? The concern is that this interpretation may set an erroneous precedent.

5) Although most of the structures presented are low-resolution, they support the relation between overall gating ring conformation and functional states. What is missing, is the relation between the different nucleotide-dependent conformations of the gating ring and actual gating of the pore. Is there a hypothesis of how the "square" gating ring stabilizes the conducting conformation, while the non-square do not? Answering these questions may be beyond the scope of the present manuscript, but some comments on these aspects may add perspective to this work.

---

## [Author Response]

Essential revisions:1) We would like to see flux assays performed on the R16A mutant.First, you showed that E125A does not assume a square conformation in any of the solved structures, and then showed that E125A is also defective in gating in the flux assay. This is all consistent with the presented gating model. It would be good to show that R16A can still gate the channel because it can adopt both square and non-square conformations. That would strengthen the proposed correlation. Second, you went on to show that E125 coordinates a Mg^2+^ ion, and that the Mg^2+^ ion is required for gating. Although R16 was also found to coordinate the same Mg^2+^ ion, the R16A structure shows that both Mg and ATP are still bound. R16A should be functional based on these observations.

We have performed flux assays with R16A, both in the sorbitol and choline chloride conditions, as requested. In line with our proposal, R16A KtrA behaves like WT KtrA and is activated by Mg^2+^.

Data are presented in Figures 6 and 8.

2) Mg^2+^ binding affinity should be better quantitated.The estimation of the apparent affinity of KtrAB for Mg^2+^ is confusing. You conclude that the K_D_of the channel for Mg^2+^ is ~1 nM, but several arguments seem incompatible with that claim. First, in 5 mM EDTA+6 mM Mg^2+^ (free [Mg^2+^]=~ 1 mM) the initial rate seems submaximal (Figure 7D, purple curve). Second, in your calculations you use a K_D_ of ~2 nM for the Mg-EDTA complex (subsection “Determination of KtrAB binding affinity for Mg^2+^”). However, this is the K_D_ of the completely deprotonated form of Mg-EDTA. The apparent K_D_ is strongly pH dependent, at pH=8.0 it is ~0.3-0.5 uM. Maxchelator or other free software should be used to estimate the apparent K_D_ of Mg-EDTA under your experimental conditions. Third, the origin of the equations used for calculating the apparent affinity of the KtrAB complex for Mg^2+^ is unclear, and needs explanation. Fourth, you claim that the estimate of a K_D_ ~1 nM was obtained assuming 2 mM contaminating Mg^2+^ in the buffer. However, in the presence of 2 mM Mg^2+^ plus 5 mM EDTA (pH=8.0) the free [Mg^2+^] is ~250 nM (Maxchelator), and under those conditions the flux rate is ~half-maximal (Figure 7E).The manuscript would be greatly improved if you could provide a more reliable estimate of the affinity of KtrAB for Mg^2+^. One straightforward approach would be to measure the total [Mg^2+^] in their 150 mM choline chloride buffer (e.g., using Mg-Green fluorescence). With that information at hand, free [Mg^2+^] could be calculated for each of the tested experimental conditions using Maxchelator. The plot in Figure 7E could then be replaced by a plot which displays initial flux rates as a function of free [Mg^2+^] (rather than EDTA), and fitted to a simple hyperbolic binding equation to obtain the apparent K_D_. Alternatively, ITC could be used to measure divalent cation binding affinity of KtrA in the presence of ATP and ADP.

We have determined the apparent affinity of Mg^2+^ by performing a magnesium ion titration using the flux assay. In this assay, we do not include EDTA in the assay solution. In addition, using flame ionization spectroscopy (after spinning down the proteoliposomes), we verified that the free-concentration of Mg^2+^ is very similar to the total-concentration of Mg^2+^ added in the assay solution. The apparent K_1/2_ of activation for Mg^2+^ is ~150 µM. These data are presented in Figure 10.

The EDTA titration experiment presented in the original version of the manuscript was therefore removed from the manuscript.

3) Quantitation of the flux assays needs revision.In principle, the steady-state fluorescence should be independent of the initial flux rate, and should reflect the fraction of liposomes that lack a reconstituted (active) KtrAB complex. A comment on the possible reasons for the strong correlation between the observed initial flux rates and the final steady-state fluorescence levels (faster initial fluxes tend to be associated with lower steady-state fluorescences) would be appreciated. Given the variable levels of steady-state fluorescence it seems inappropriate to report flux rates as initial slopes, since the initial slope depends not only on the flux rate, but also on the total amplitude of the signal (F_initial_ – F_steady-state_). One way to obtain more reliable flux rates would be to fit the fluorescence time courses with single decaying exponentials and report the fitted rate constants. Alternatively, the "normalized fluorescence" signal could be redefined as NF=(F – F_ss_)/(F_ini_ – F_ss_), where F_ss_ is the steady-state fluorescence (before valinomycin addition). This transformation would equalize the total amplitude of each signal and would make comparison of initial slopes more meaningful.

We thank the reviewers for pointing-out the glaring flaw in our analysis of the flux assay curves. We have now re-analyzed all the data using exponential fits and present the rate constants extracted from the fits to understand the activation mechanism. Importantly, the global conclusions of the manuscript have not changed.

We have decided to continue to present the fluorescence quenching curves normalized with the valinomycin values. However, the data collected in the cation titrations were normalized using the procedure suggested by the reviewers (Figure 10). In addition, we provide a supplement figure (Figure 10—figure supplement 1) that shows the fluorescence raw data and “valinomycin normalized” data for the Mg^2+^ titration in an effort to explain the variations in the steady-state fluorescence levels between curves.

4) Subsection “Magnesium binding site”: "instead of water molecules, nitrogen atoms of the R16 guanidinium group appear to complete the coordination shell of the divalent cation"Are there any other known examples of an arginine side chain coordinating a divalent cation? Mg does not have the character of a transition metal, for which one might expect coordination by N atoms, and the N-Mg distances shown in Figure 5D are much longer than what would be expected for direct coordination. Is it possible that there are water molecules present in the coordination shell whose density is weak or unresolved, and that there is uncertainty in the orientation of the R16 guandinium in this 3.2 A structure? The concern is that this interpretation may set an erroneous precedent.

This is a good point. We now mention in the text that there are high-resolution structures displaying coordination of Mg^2+^ by guanidium groups of arginines and indicate their PDB codes. One of these structures shows a similar interaction distance.

5) Although most of the structures presented are low-resolution, they support the relation between overall gating ring conformation and functional states. What is missing, is the relation between the different nucleotide-dependent conformations of the gating ring and actual gating of the pore. Is there a hypothesis of how the "square" gating ring stabilizes the conducting conformation, while the non-square do not? Answering these questions may be beyond the scope of the present manuscript, but some comments on these aspects may add perspective to this work.

We have added a paragraph to the Discussion section where we address this point.